# Synthesis and Characterization of Gefitinib and Paclitaxel Mono and Dual Drug-Loaded Blood Cockle Shells (*Anadara granosa*)-Derived Aragonite CaCO_3_ Nanoparticles

**DOI:** 10.3390/nano11081988

**Published:** 2021-08-02

**Authors:** S. Chemmalar, Abdul Razak Intan-Shameha, Che Azurahanim Che Abdullah, Nor Asma Ab Razak, Loqman Mohamad Yusof, Mokrish Ajat, N. S. K. Gowthaman, Md Zuki Abu Bakar

**Affiliations:** 1Natural Medicines and Products Research Laboratory, Institute of Bioscience, Universiti Putra Malaysia (UPM), Serdang 43400, Selangor, Malaysia; gs52461@student.upm.edu.my (S.C.); norasmarazak@upm.edu.my (N.A.A.R.); mokrish@upm.edu.my (M.A.); zuki@upm.edu.my (M.Z.A.B.); 2Department of Veterinary Preclinical Sciences, Faculty of Veterinary Medicine, Universiti Putra Malaysia (UPM), Serdang 43400, Selangor, Malaysia; 3Biophysics Laboratory, Department of Physics, Faculty of Science, Universiti Putra Malaysia (UPM), Serdang 43400, Selangor, Malaysia; azurahanim@upm.edu.my; 4Material Synthesis & Characterization Laboratory, Institute of Advanced Technology, Universiti Putra Malaysia (UPM), Serdang 43400, Selangor, Malaysia; 5Laboratory of Cancer Research (MAKNA), Institute of Bioscience, Universiti Putra Malaysia (UPM), Serdang 43400, Selangor, Malaysia; 6Department of Companion Animal Medicine and Surgery, Faculty of Veterinary Medicine, Universiti Putra Malaysia (UPM), Serdang 43400, Selangor, Malaysia; loqman@upm.edu.my; 7School of Engineering, Monash University Malaysia, Jalan Lagoon Selatan, Bandar Sunway, Subang Jaya 47500, Selangor, Malaysia; gowthaman.nalla@monash.edu

**Keywords:** calcium carbonate nanoparticles, blood cockle shells, mesoporous, gefitinib, paclitaxel, mono drug loading, dual drug loading, XRD, FTIR

## Abstract

Calcium carbonate has slowly paved its way into the field of nanomaterial research due to its inherent properties: biocompatibility, pH-sensitivity, and slow biodegradability. In our efforts to synthesize calcium carbonate nanoparticles (CSCaCO_3_NP) from blood cockle shells (*Anadara granosa*), we developed a simple method to synthesize CSCaCO_3_NP, and loaded them with gefitinib (GEF) and paclitaxel (PTXL) to produce mono drug-loaded GEF-CSCaCO_3_NP, PTXL-CSCaCO_3_NP, and dual drug-loaded GEF-PTXL-CSCaCO_3_NP without usage of toxic chemicals. Fourier-transform infrared spectroscopy (FTIR) results reveal that the drugs are bound to CSCaCO_3_NP. Scanning electron microscopy studies reveal that the CSCaCO_3_NP, GEF-CSCaCO_3_NP, PTXL-CSCaCO_3_NP, and GEF-PTXL-CSCaCO_3_NP are almost spherical nanoparticles, with a diameter of 63.9 ± 22.3, 83.9 ± 28.2, 78.2 ± 26.4, and 87.2 ± 26.7 (nm), respectively. Dynamic light scattering (DLS) and N_2_ adsorption-desorption experiments revealed that the synthesized nanoparticles are negatively charged and mesoporous, with surface areas ranging from ~8 to 10 (m^2^/g). Powder X-ray diffraction (PXRD) confirms that the synthesized nanoparticles are aragonite. The CSCaCO_3_NP show excellent alkalinization property in plasma simulating conditions and greater solubility in a moderately acidic pH medium. The release of drugs from the nanoparticles showed zero order kinetics with a slow and sustained release. Therefore, the physico-chemical characteristics and in vitro findings suggest that the drug loaded CSCaCO_3_NP represent a promising drug delivery system to deliver GEF and PTXL against breast cancer.

## 1. Introduction

Breast cancer accounts for almost one in four cancer cases in women worldwide [1]. According to the Global Cancer Observatory (GLOBOCAN), in the year 2020, the incidence of breast cancer in women was 2,261,419 (11.7%) and the mortality was 684,996 (6.9%) [2]. The molecular mechanisms of breast cancer are complex, but three main classes of breast cancer have been identified, namely progesterone positive (PR+) or estrogen receptor-positive (ER+), human epidermal receptor-positive (HER+), and triple negative breast cancer (TNBC), where TNBC lacks the ER, PR, and HER [3]. Epidermal growth factor receptor (EGFR) is a transmembrane protein with cytoplasmic kinase activity and its overexpression has been detected in 30% to 60% of human breast cancer biopsies, being indicative of a poor prognosis [4]. Overexpression of the HER 2 receptor has been noticed in a significant number of cases of breast cancer [5], and it activates ligand-dependent stimulation of the kinase domain, resulting in the unprompted formation of dimers. HER 2 receptors’ capacities to potentiate signaling of HER 1 create potentiated growth stimuli that lead to stimulation of further intracellular pathways, which in turn maintain the explosive cell proliferation rates associated with the tumor. Hence, a rational choice of treatment for EGFR-expressing breast cancers would be to inhibit the EGFR-driven autocrine pathway. AstraZeneca’s ZD 1839, is a low-molecular-weight (446.9) synthetic anilinoquinazoline, also called gefitinib or Iressa^®^ [6,7], is a great choice for incorporating into chemotherapy since it is a selective, reversible inhibitor of EGFR tyrosine kinase [8,9]. Gefitinib (GEF) is structurally similar to ATP and has a higher affinity to ATP site in the clefts of the EGFR than ATP itself [7]. The inhibition of kinase has several effects, including cell cycle arrest, induction of apoptosis, inhibition of invasion and metastasis of tumor cells, and finally, augmentation of the antitumor effects of chemotherapy/radiation [10].

Conventional chemotherapy has been used in the clinical setting for many decades, although it poses the problems of drugs simply diffusing after they are administered and spreading freely throughout the body. This exposes healthy cell to the drugs, and results in obnoxious side-effects. Besides, only a small amount of the dosage administered reaches the actual tumor to elicit a therapeutic response [11,12]. Over the past decades, there has been a shift in focus towards nanotechnology and nanomedicine for cancer therapy since nanomedicines can overcome the lack of specificity of conventional chemotherapeutic agents, reduce the overall systemic toxicities, and help in delivering the payload to the tumor site safely. The dogma behind nanomedicines against cancer is the enhanced permeability retention effect (EPR). This is because tumors have leaky vasculature due to the sustained tumor angiogenesis and the tumor contains underdeveloped lymphatic drainage [13], which facilitates the passive directing of nanoparticles to the tumor [12]. Calcium carbonate is one of the inorganic compounds which worked its way into the field of nanomaterial research due to its inherent properties of availability, biocompatibility, pH-sensitivity, and slow biodegradability [14,15,16]. Calcium carbonate nanoparticles decompose slowly at the normal physiological pH (7.4) while it displays a faster decomposition in the acidic pH (<6.5) typical of the tumor environment [17]. Tumor cell death was achieved without any toxicity with the administration of calcium carbonate nanoparticles [18]. De novo synthesized CaCO_3_ nanoparticles alkalinized the tumor microenvironment and shrank the tumor [19]. One of Natures’ well-known reserves of calcium carbonate are seashells. The shells are formed due to the biomineralization process and are 95–99% comprised of calcium carbonate [20,21,22]. Calcium carbonate can be of six types, where the aragonite and calcite forms are the most widely produced in a biological setup [23]. The present study aims at developing an eco-friendly and biocompatible nanoparticulate anticancer drug from the shells of blood cockles (*Anadara granosa*), a marine bivalve organism that grows along the shallow waters of the coastline surrounding Malaysia. Blood cockles are a staple in the diet of Malaysians and millions of tons of shell food waste is currently being dumped into the environment, due to the increase of shellfish aquaculture. Recycling shell waste in various avenues will form the best solution to overcome the environmental pollution issue and improve sustainable living [24]. Shell wastes have been utilized in various applications such as in wastewater decontamination, as a soil conditioner, feed additive [25,26], and as a liming agent in construction.

Drug delivery systems where drugs or active principles are loaded into nanocarriers are very different from conventional drug delivery systems. Combination therapy is the current trend for cancer treatment, and the usage of nanomedicines in clinical settings is still in its infancy. Combining both these technologies will yield better chemotherapeutic systems, serving as the “magic pill” for cancer patients. The mode of action and toxicity profile of gefitinib compared to traditional cytotoxic drugs and radiation therapy provided an ideal rationale for using is in combination with conventional cytotoxic drugs to achieve additive or synergistic anticancer effects [27]. Combination therapy came into use in chemotherapy because of various factors that include overcoming drug resistance and multitargeted treatments for disturbing multiple nodes of pathways of interest for better treatment outcomes [28]. Synergistic and potentiated drug combinations like GEF and paclitaxel (PTXL) will help achieve one or more promising outcomes: improved efficacy, decreased dosage, and delayed development of drug resistance. Pharmacodynamically synergistic drug combination due to anti-counteractive actions—different targets of cross-talking pathways have been observed in different breast cancer cell lines. It was found that GEF produces anticancer effects, which assists the counteractive EGFR-hypoxia crosstalk in resisting PTXL pro-apoptotic property [29]. In vitro and in vivo studies with various cancer cell lines [30], has shown synergism between GEF and PTXL [31]. Considering all the beneficial properties of CSCaCO_3_NP and the synergistic relationship of GEF and PTXL, this study was designed to load GEF and PTXL into CSCaCO_3_NP to be used for intra-tumoral or peri-tumoral administration against breast cancer. This is the one-of-a-kind study where this drug combination is loaded on pH-dependent CSCaCO_3_NP which would serve not only as a carrier but also as a local alkalizer to bring down the pH of the tumor microenvironment. Since higher hydrogen ion concentration in the tumor microenvironment has been correlated to increased tumor growth, local invasion, and metastasis [32], buffering of the tumor microenvironment has pathological effects on tumor cells. Moreover, PTXL and GEF are hydrophobic drugs and PTXL requires a solubilizer such as Cremophor EL which adds to the adverse effects faced by the patients. GEF needs to be taken per-os every 24 h. Hence designing a drug system which will carry the payload to the tumor site without any loss will mitigate the adverse effects caused by conventional chemotherapy. The biogenic calcium carbonate nanoparticles derived from the blood cockle shells have shown promising results in delivering chemotherapeutic drugs [14,15,33]. Various drug-loaded calcium carbonate nanoparticles have been synthesized with hydrophilic drugs [33,34,35,36], hydrophobic drugs [37], bioactive proteins [36], hormones [38], antibiotics [39,40], nucleic acids [41] to hybrids of alginate/CaCO_3_ with DOX and PTXL [42], and heparin/CaCO_3_/CaP nanoparticles [43]. Conventional drugs with lower solubility such as GEF and PTXL when loaded onto the pH-dependent CSCaCO_3_NP, characterization becomes vital in the path of new drug development. Adequate testing as per the National Characterization Laboratory (NCL) guidelines must be followed [44].

For the first time, we have synthesized calcium carbonate nanoparticles derived from shells of the blood cockles using Tween 80 and mechanical dry ball milling. We also have developed a method to load the drugs GEF and PTXL onto CSCaCO_3_NP to synthesize the novel mono drug-loaded GEF-CSCaCO_3_NP and PTXL-CSCaCO_3_NP and novel dual drug-loaded GEF-PTXL-CSCaCO_3_NP. To quantify the concentration of drug that is loaded onto the nanoparticles, a simple UV-Vis spectrophotometry assay was developed. A method for the simultaneous estimation of GEF and PTXL was developed and validated for the first time. We have investigated the various physicochemical characteristics of the synthesized nanoparticles. The alkalizing and solubilizing properties of CSCaCO_3_NP and the drug release kinetics of GEF-CSCaCO_3_NP, PTXL-CSCaCO_3_NP, and GEF-PTXL-CSCaCO_3_NP in various pH were determined. Also, we have demonstrated that GEF and PTXL can be loaded onto the inorganic CSCaCO_3_NP, which have unique and advantageous physicochemical properties suitable to be used against breast cancer in future studies.

## 2. Materials and Methods

### 2.1. Materials and Reagents

GEF and PTXL (Gold Biotechnology, St. Louis, MO, USA), DMSO (Fisher Scientific U.K. Limited, Loughborough, UK), Tween 80 (R & M Marketing, Essex, UK), deionized water of HPLC grade with a resistance of 18.2 cm MΩ was obtained from a Milli-Q integral Water Purification System (Type I) for Ultrapure water (Merck KGaA, Darmstadt, Germany), double beam UV-Vis spectrophotometer (1650PC, Shimadzu Europe, Duisburg, Germany), high-speed centrifuge (Optima XPN, Beckman Coulter Life Sciences, Indianapolis, IN, USA), magnetic stirrer (WiseStir^®^ Systematic Multi-Hotplate Stirrer, Dhaihan Scientific^®^, Gangwon-do, Korea), hot air oven (UM500, Memmert GmbH+ Co.KG, Schwabach Germany), programmable ball miller (BML-6″, Diahan Scientific^®^), filter paper (Filtres Fioroni, Ingré, France), Field emission scanning electron microscope (Nova^TM^NanoSEM 230, FEI, Hillsboro, OR, USA), carbon-coated copper grid (Sigma-Aldrich, St. Louis, MO, USA), Energy dispersion X-ray (EDX) spectroscopy (X-max EDS, Oxford Instruments, Abingdon, UK), Zetasizer Nano ZS (Ver.7.11; Malvern Instruments Ltd., Malvern, UK), XRD (Shimadzu XRD-6000 powder diffractometer), FT-IR (Spectrum 100; Perkin Elmer, Shelton, CT, USA), Tristar II Plus (Micromeritics, Norcross, GA, USA), dialysis tubing cellulose membrane (typical molecular weight cut-off = 14,000, Sigma Aldrich, Merck KGaA, Darmstadt, Germany), shaker incubator (Hybaid Maxi 14, Thermo Fischer Scientific, Guldensporenpark, Merelbeke, Belgium) pH meter (LAQUA PH1100, Horiba, Minami-ku, Kyoto, Japan), nitric acid 65% for analysis EMSURE^®^ (Merck KGaA, Darmstadt, Germany), hydrogen peroxide 35% (R & M Marketing, Essex, UK), PBS tablets (MP Biomedicals LLC, Santa Ana, CA, USA), Albumin from bovine serum (>96% GE, 05473, Fluka, Merck KGaA, Darmstadt, Germany ), Atomic absorption spectrometer (900T, Perkin Elmer, Waltham, MA, USA). *Anadara granosa* Linnaeus, 1758 (Mollusca, Bivalvia, Arcidae) were sourced from the local market in Serdang, Malaysia.

### 2.2. Synthesis of CSCaCO_3_NP, GEF-CSCaCO_3_NP, PTXL-CSCaCO_3_NP, and GEF-PTXL-CSCaCO_3_NP

#### 2.2.1. Synthesis of CSCaCO_3_NP from Cockle Shells Using Tween 80 and Ball Milling

The synthesis of CSCaCO_3_NP comprises of two steps (Steps 1 and 2 of Figure 1). The first step was the synthesis of micron aragonite calcium carbonate (MAC) powder from blood cockle shells (Step 1 of Figure 1) [45]. The synthesis of nanoparticles is as per the step 2 of Figure 1. It entailed placing 2 g of the 75-µm of CSCaCO_3_ powder and the addition of 20 mL of deionized water. The concoction was sonicated for fifteen minutes. Then 1 mL of Tween 80 was added, and the suspension was stirred at 1100 rpm with a magnetic stirrer bar for 2 h at room temperature. The obtained suspension was filtered using filter paper. The filtrate was centrifuged for 10 min at 14,000 rpm, and the supernatant was discarded. The pellet was dispersed upon the addition of 20 mL deionized water and then washed twice with deionized water. The collected nanoparticles were dried in an oven at 50 °C for 48 h. Upon drying, the nanoparticle was added to the dry milling container (8 × 11 cm). Milling was carried out using a programmable ball miller with a 10:1 ball to powder ratio, with ceramic balls (eight balls of 1.6 cm diameter and two balls of 2.5 cm diameter). This jar was placed on a programmable ball miller and operated at 120 rpm for 120 h. The resultant CSCaCO_3_NP was then stored in an oven at 50 °C.

#### 2.2.2. Synthesis of GEF-CSCaCO_3_NP, PTXL-CSCaCO_3_NP, and GEF-PTXL-CSCaCO_3_NP

GEF-CSCaCO_3_NP, PTXL-CSCaCO_3_NP, and GEF-PTXL-CSCaCO_3_NP were prepared by the following method (Step 3 of Figure 1). Three groups were devised to determine the perfect combination of drugs and CSCaCO_3_NP. The experimental grouping is shown in Table 1. CSCaCO_3_NP suspension was prepared with 0.05% Tween 80 buffer and DMSO in a 50:50 ratio. As per the groups, the drug dissolved in DMSO was added into the suspension drop by drop. The loading of drugs was achieved by continuous stirring at 200 rpm overnight at room temperature in a dark environment. The individual suspensions were centrifuged at 14,000 RPM for 10 min, followed by washing the pellet with deionized water and drying. The supernatants of mono and dual-drug loaded nanoparticles were collected and measured using a UV-Vis spectrophotometer to determine the amount of un-entrapped GEF and PTXL. All experiments were carried out in triplicate.

### 2.3. Physicochemical Characterization of CSCaCO_3_NP, GEF-CSCaCO_3_NP, PTXL-CSCaCO_3_NP and GEF-PTXL-CSCaCO_3_NP

#### 2.3.1. UV-Vis Spectrophotometry

Absorbance measurements were made using a double beam ultraviolet-visible spectrophotometer (Shimadzu 1650PC) at the wavelength range: 200–400 nm at a scan speed of 0.1 nm and with a constant slit width of 2.0 nm. The stock solution of GEF (20 mg/mL) was prepared by adding 10 mL of DMSO to 200 mg of GEF and sonicating for 5 min to ensure complete solubilization. The stock solution of PTXL (10 mg/mL) was prepared by adding 10 mL of DMSO to 100 mg of PTXL and sonicating for 5 min to ensure complete solubilization. DMSO was used without any further dilution. Tween 80 buffer was prepared by adding 10 µL of Tween 80 to 20 mL deionized water and sonicating for 5 min. A combination of DMSO and Tween 80 buffer (50:50) was used as the solvent in all the experiments. The stock solutions were measured and added to the solvent mixture such that the final concentration is 2.5–25 µg/mL for GEF and 10–90 µg/mL for PTXL, were obtained. In the dual drug system, 2.5–20 µg/mL for GEF and 1.25–10 µg/mL for PTXL were mixed in various concentrations, but the ratio of 2:1 was maintained for GEF: PTXL throughout. The blank used for these measurements was the solvent mixture without the drug. The working standard solutions were mixed and scanned in the UV range from 200 nm through 400 nm to determine the absorption maxima (λ_max_). The sample volume used was 4 mL, and measurements were taken at room temperature. Various concentrations of GEF, PTXL, and GEF+PTXL were assayed. The absorbances of the resulting solutions were measured at 332 and 248 nm for GEF and PTXL. For the dual drug admixture, GEF and PTXL were measured at 332 and 248 nm. Drug interaction studies were carried out. The procedure was repeated thrice, and the mean absorbances were calculated. The concentrations obeying the Beer-lambert law were found and fitted for calibration with the OriginPro 9.0 program (OriginLab Corporation, Northampton, MA 01060, USA). The amount of GEF and PTXL loaded onto the calcium carbonate nanoparticles was determined by measuring the absorbance of the resulting supernatant obtained after centrifugation following the loading of drugs (From method 2.2.2). The unknown concentration was determined using the OriginPro 9.0 program. The entrapment efficiency (EE%) and loading content (LC%) was determined as per the formula [38]. The method was validated according to the International Conference on Harmonization Q2(R1) guidelines [46]. The analytical performance criteria which were tested for this method validation: linearity, recovery (%), precision, and sensitivity.

#### 2.3.2. Field Emission Scanning Electron Microscopy (FESEM), and Energy Dispersion X-ray (EDX) Spectroscopy

For FESEM, the sample was dispersed onto 12 mm diameter aluminium sample holders using conductive carbon paint and then sputter-coated with a platinum layer under vacuum. This sample holder was then installed into the FESEM and examined. The elemental composition of the CSCaCO_3_NP, GEF, PTXL, GEF-CSCaCO_3_NP, PTXL-CSCaCO_3_NP, and GEF-PTXL-CSCaCO_3_NP was determined using Energy dispersion X-ray (EDX) spectroscopy. We analyzed the images using ImageJ software (ImageJ 1.52a, National Institutes of Health, USA), and the data was processed with GraphPad Prism 8.0.2 (San Diego, CA, USA).

#### 2.3.3. BET and BJH Analysis with N_2_ Physisorption Isotherms

For N_2_ adsorption and desorption experiments, the sample was initially outgassed at 90 °C for 60 min; Nitrogen adsorption and desorption at 77 K was carried out with a Micromeritics Tristar II Plus system. Based on the absorption and desorption isotherm at a relative pressure range of (P/P_0_) from 0.01 to 0.991 fractions, the data generated were analyzed using the Brunauer-Emmett-Teller (BET) method and Barrett-Joyner-Halenda (BJH) models. We determined the BET specific surface area and BJH mean pore size from the isotherms.

#### 2.3.4. Dynamic Light Scattering

For zeta potential (ζ-potential) and hydrodynamic size analysis, 0.4 mg of each sample of CSCaCO_3_NP, GEF-CSCaCO_3_NP, PTXL-CSCaCO_3_NP, and GEF-PTXL-CSCaCO_3_NP was dispersed in 15 mL of deionized water and sonicated for 30 min. The sample to be analyzed was again double diluted with deionized water. For analysis of pure drugs, was dissolved in 20 mL of phosphate buffered saline (PBS) of pH 7.4 with 1% Tween 80 was used to dissolve 2 mg of GEF and 1 mg of PTXL, respectively. This solution was further diluted with PBS before the measurements. The sample was injected into disposable cuvettes and placed inside the Zetasizer Ver. 7.11 Nano ZS system. The hydrodynamic diameter was obtained from the intensity distribution using software version 2.2. We carried out three independent experiments to get the average hydrodynamic diameter and zeta potentials. The polydispersity index was measured along with the hydrodynamic diameter readings.

#### 2.3.5. Powder X-ray Diffraction (PXRD) and Fourier-Transform Infrared Spectroscopy (FTIR)

PXRD patterns were obtained using the Shimadzu XRD-6000 powder diffractometer with Cu Kα (λ = 1.540562 Å) as the X-ray source. The samples were scanned at the step size of (°2 Th.) 0.033 and scan step time (s) of 19.685 with diffraction angles ranging from 4.0207° to 89.9527° a voltage of 40 kV and current of 40 mA. We analyzed the obtained PXRD data with the X’Pert HighScore Plus 2.2 program using search-match operations with the Inorganic Crystal Structure Database (ICSD). The crystallite size D (Å) was determined using the Scherrer equation. Vesta software was used for creating the crystallographic model. For FTIR, the samples were analyzed using the Fourier infrared spectrophotometer (Model 100 series, Perkin Elmer) using the KBr pellet method as described [47], over the range of 4000 to 400 cm^−1^ at a 2 cm-^1^ resolution and averaging 64 scans. We analyzed the obtained data using the KnowItAll Informatics System 2020 (Bio-Rad, John Wiley & Sons, Inc., Hoboken, NJ, USA). The chemical structures were drawn using ChemBioDraw Ultra 12.0 (CambridgeSoft, Perkin Elmer, Shelton, CT, USA).

#### 2.3.6. In Vitro Release Studies of GEF and PTXL

The in vitro drug release studies of GEF and PTXL were carried out in PBS with 0.2% (*v*/*v*) Tween 80 at pH 6.5, 5.6, and 7.4 at 37 °C by the dialysis method [48]. Two mL of PBS containing GEF-CSCaCO_3_NP, PTXL-CSCaCO_3_NP, and GEF-PTXL CaCO_3_NP containing 800 µg, 800 µg and 800 µg + 400 µg of GEF, PTXL, and GEF+PTXL, respectively, was pipetted into the pre-treated dialysis membrane and suspended in 15 mL of PBS. The tubes were kept inside a thermostatic shaker (Hybaid Maxi 14) at 37 °C at 100 rpm. At pre-determined intervals, 2.5 mL of supernatant was withdrawn and the concentrations of GEF and PTXL were measured using a double beam UV-Vis spectrophotometer. At scheduled time intervals of 0, 2, 6, 24, 46, 58, 72, 86, and 100 h, 2.5 mL of the release media was taken out and the same volume of fresh buffer was replaced. The absorbances of drugs present in the release media was determined using UV-Vis Spectrophotometer and the unknown concentrations were calculated using the OriginPro 9.0 program. The release experiments were conducted in triplicates for the various samples and the results were expressed as mean value ± SD. The results were presented as cumulative drug release percentages (%) against time (hours). The in vitro drug release data were fitted in various kinetic models such as zero order, first order, Krosmeyer-Peppas and Higuchi to determine the appropriate drug release mechanism.

#### 2.3.7. Dissolution Analysis of CSCaCO_3_NP

The dissolution of CSCaCO_3_NP were measured by dispensing 60 mg of CSCaCO_3_NP in 30 mL of with 0.2% (*v*/*v*) Tween 80 phosphate buffered saline and simulated plasma conditions with pH of 7.4, 6.5 and 5.6. The plasma simulating fluid was prepared by adding 5 g of albumin to 100 mL phosphate buffered saline These samples were incubated at 37 °C, stirred at 100 rpm in a thermostatic shaker (Hybaid Maxi 14). At designated time, 9 mL of supernatants were collected after ultracentrifugation at 14,000 RPM for 10 min. 9 mL of the respective media was replaced to the tubes. The collected supernatants were predigested by addition of 8 mL of nitric acid and 1 mL of 33% a Hydrogen peroxide and heated at ~160 °C, till the mixture is completely digested and clear. The total calcium contents were measured by atomic absorption spectrometry (Perkin Elmer 900T) at the wavelength of 422.67 nm. The analysis was repeated thrice, and the calcium concentration was expressed as mean ± SD.

#### 2.3.8. Alkalinizing Property of CSCaCO_3_NP

The alkalinizing profiles of CSCaCO_3_NP were measured in simulated plasma fluid and phosphate buffered saline with 0.2% (*v*/*v*) Tween 80. The plasma simulated fluid and Tween 80 buffered PBS were adjusted to three pH of 7.4, 6.5 and 5.6. The alkalizing property of CSCaCO_3_NP was measured by dispersing the nanoparticles at three different concentrations of CSCaCO_3_NP of 5 mg/mL, 1.66 mg/mL, and 0.83 mg/mL in plasma simulated fluid and Tween 80 buffered PBS, respectively. These samples were incubated at 37 °C and stirred at 100 rpm in a thermostatic shaker (Hybaid Maxi 14), and at designated time intervals for 24 h, the pH was measured using the pH meter (LAQUA PH1100). The analysis was repeated thrice, and the pH was expressed as mean ± SD.

## 3. Results

Physicochemical characterization is required to determine the following characteristics: size, shape, homogeneity, surface area, surface charge, degree of dispersion, drug loading capacity, surface functional groups, and phase of the nanoparticles. These characteristics are essential to determine their biological interaction and kinetics of the nanoparticles in the biological milieu.

### 3.1. UV-Vis Spectrophotometry, Drug Loading Content and Encapsulation Efficiency

A precise method based on UV-vis spectroscopy for the quantification of GEF and PTXL in a CSCaCO_3_NP formulation has been developed. GEF and PTXL in a mixture of an equal volume of DMSO and Tween 80 buffer (0.05%) produced a characteristic curve in the UV wavelength range between 200 and 400 nm. The λ_max_ of 332.6 ± 0.2 nm and 247.8 ± 0.3 nm was satisfactory for GEF and PTXL, and for the dual drugs, the λ_max_ of 332.5 ± 0.1 nm for GEF and 248.3 ± 0.8 nm for PTXL was used (Figure A1a). The absorbances of the resulting solutions were measured at 332 and 248 nm for GEF and PTXL (Table A1). For the dual drug admixture, GEF and PTXL were measured at 332 and 248 nm (Table A2). Drug interaction spectra is shown in Figure A1. The calibration curves plotted with the six points (Figure A1b). The drug interaction between GEF and PTXL at various concentrations is shown in Figure A1c,d. The individual concentration range obeying Beer-lambert law, and the regression characteristics are given in Table A3. The data obtained for analytical performance criteria for this method validation were Linearity (Table A2 and Table A3), Recovery and precision (Table A4 and Table A5) and sensitivity (Table A5).

For the drug loading, the approach was a simple stirring method for the physical adsorption of drugs, which was facile and efficient. Since GEF and PTXL are hydrophobic drugs, DMSO with Tween 80 buffer (0.05%) was the solvent used for the loading process. The supernatant obtained after centrifugation was examined for the peaks using the UV-Vis Spectrophotometer, and it had remained unchanged compared to the spectra obtained from the pure drugs. The unaltered spectra indicate that no chemical changes have occurred during the loading procedure. No interference arose from the CSCaCO_3_NP during the measurement of the absorbances of the drug in the supernatant.

The loading content is the capacity of the CSCaCO_3_NP to load the drugs, the amount of total entrapped drug per total weight of the nanoparticles. The entrapment efficiency is the drug’s ability to be encapsulated efficiently, and it gives the percentage of the drug entrapped successfully. Loading capacity and the entrapment efficiencies of GEF and PTXL in various groups of the mono and dual drug-loaded CSCaCO_3_NP are given in Figure 2a,b.

### 3.2. Field Emission Scanning Electron Microscopy (FESEM), and Energy Dispersion X-ray (EDX) Spectroscopy

The FESEM micrographs of CSCaCO_3_NP revealed particles with an average size of 63.9 ± 22.3 nm (Figure 3). GEF-CSCaCO_3_NP, PTXL-CSCaCO_3_NP and GEF-PTXL-CSCaCO_3_NP exhibited spherical shape and uniform size of 83.9 ± 28.2, 78.2 ± 26.4, and 87.2 ± 26.7 nm, respectively (Figure 3).

The elemental composition of CSCaCO_3_NP, GEF-CSCaCO_3_NP, PTXL-CSCaCO_3_NP, GEF-PTXL-CSCaCO_3_NP, GEF and PTXL are shown in Table 2. The EDX data show that calcium, carbon, and oxygen are the major elements of the CSCaCO_3_NP and sodium is present in a very meagre amount. The Ca/C/O are in a molar ratio close 1:1:3 as per the conventional stoichiometrically-derived ratio of CaCO_3_.

### 3.3. Specific Surface Area and Pore Size

From the nitrogen adsorption and desorption experiments, the resulting isotherm for CSCaCO_3_NP, GEF-CSCaCO_3_NP, PTXL-CSCaCO_3_NP, and the GEF-PTXL-CSCaCO_3_NP is a Type IV isotherm (Figure 3). The hysteresis loop is formed in all the isotherms. This hysteresis loop shows that the adsorption and desorption curves are almost vertical and parallel, and it belongs to Type 1 and 3. This type of geometry in the isotherm is indicative of mesoporous pore type. The hysteresis loop of CSCaCO_3_NP begins at ~0.62 P/P^0^ and ended closer to 1. For the drug loaded nanoparticles, the hysteresis loop began at ~0.82 to ~0.86 and ended very close to the value of 1. GEF-CSCaCO_3_NP, PTXL-CSCaCO_3_NP, and GEF-PTXL-CSCaCO_3_NP. The surface area of the synthesized CSCaCO_3_NP, GEF-CSCaCO_3_NP, PTXL-CSCaCO_3_NP, and GEF-PTXL-CSCaCO_3_NP are 10.7, 8.3, 9.4, and 9.9 (m^2^/g), respectively. The BJH mean pore width of the CSCaCO_3_NP, GEF-CSCaCO_3_NP, PTXL-CSCaCO_3_NP, and GEF-PTXL-CSCaCO_3_NP were 5.2, 5.4, 5.6, and 5.2 (nm), respectively.

### 3.4. Dynamic Light Scattering

The Zeta (ζ)-potential (mV), hydrodynamic diameters (nm), and PDI of the synthesized CSCaCO_3_NP, GEF-CSCaCO_3_NP, PTXL-CSCaCO_3_NP and GEF-PTXL-CSCaCO_3_NP was obtained (Table 3).

### 3.5. Powder X-ray Diffraction

Figure 4a shows the X-ray diffraction patterns obtained for the investigated samples: CSCaCO_3_NP, GEF-CSCaCO_3_NP, PTXL-CSCaCO_3_NP and GEF-PTXL-CSCaCO_3_NP. One can observe that the diffraction peaks of the drugs GEF and PTXL are invariably different from the diffraction peaks obtained from the nanoparticle samples. In the Inorganic Crystal Structure Database (I.C.S.D.), the samples were matched to the ICSD patterns with the highest score: 98-006-2988, 98-010-9087, 98-006-2988, and 98-001-6937 for CSCaCO_3_NP, GEF-CSCaCO_3_NP, PTXL-CSCaCO_3_NP, and GEF-PTXL-CSCaCO_3_NP, respectively. The crystal structure obtained from the synthesized CSCaCO_3_NP (Figure 4b). The lattice parameters are also shown in the same figure. The crystallite size (Å) of CSCaCO_3_NP, GEF-CSCaCO_3_NP, PTXL-CSCaCO_3_NP, and GEF-PTXL-CSCaCO_3_NP was 328, 319, 301, and 328, respectively. The diffractograms of GEF and PTXL are shown in Figure A2.

### 3.6. Fourier-Transform Infrared Spectroscopy (FTIR)

The FTIR spectra of CSCaCO_3_NP showed vibrational bands at 1445, 1084, 856, and 714 cm^−1^ (Figure 5). The largest and strongest band appeared at 1445 cm^−1^, and smaller peaks at 1084 and 856 cm^−1^ are attributed to CO_3_^2−^ in the CaCO_3_ molecular structure. The GEF-PTXL-CSCaCO_3_NP showed new spectral absorption peaks at 952.84 (cyclohexane), 1024.20 (C-F stretch), 2918.30 (C-H stretching) and 3435.22 (aromatic amine and OH^−^ stretch) (Figure 5). The largest and strongest band exhibited by CSCaCO_3_NP at 1445 cm^−1^ remained unaltered in the spectra of GEF-PTXL-CSCaCO_3_NP, GEF-CSCaCO_3_NP and PTXL-CSCaCO_3_NP, indicating that the alkyl group is unaffected. The spectral absorption peaks of GEF-CSCaCO_3_NP displayed less pronounced 1024.20 (C-F stretch) and 952.84 (cyclohexane) cm^−1^ peaks as compared to GEF-PTXL-CSCaCO_3_NP. The spectral absorption peaks of PTXL-CSCaCO_3_NP showed a slightly more pronounced 952.84 cm^−1^ (cyclohexane) peak than GEF-PTXL-CSCaCO_3_NP (Figure 5). The FTIR spectra of GEF and PTXL revealed many absorption peaks corresponding to the chemical groups present in their structures (Figure A3a,b).

### 3.7. In Vitro Drug Release

In vitro release of GEF and PTXL from GEF-CSCaCO_3_NP, PTXL-CSCaCO_3_NP, and GEF-PTXL-CSCaCO_3_NP were studied in PBS containing 0.2% Tween 80 at various pH of 7.4, 6.5 and 5.6 at 37 °C for 100 h. In all the formulations, the release of GEF and PTXL did not exceed more than 30% at 100 h, indicating that the drugs have a slow and sustained release (Figure 6a–e). The data also indicate that the drugs are released at a higher concentration in the moderately acidic pH of 5.6, when compared to the release of the drugs at the pH values of 6.5 and 7.4. The cumulative release of GEF at the end of 100 h was ~16, ~17 and ~19% from GEF-CSCaCO_3_NP in the pH of 7.4, 6.5 and 5.6, respectively. The cumulative release of PTXL at the end of 100 h was ~18, ~21 and ~22% from PTXL-CSCaCO_3_NP in the pH of 7.4, 6.5 and 5.6, respectively. The cumulative release of GEF/PTXL at the end of 100 h was ~22/18, ~23/20 and ~26/23% from GEF-PTXL-CSCaCO_3_NP in the pH of 7.4, 6.5 and 5.6, respectively. The release characteristics could be attributed to the pH dependent solubility of the nanocarrier.

The experimental release data for GEF and PTXL in the three formulations were fitted to different kinetic models. The best fitting model with the R^2^ value (close to 1) obtained for different models (Table 4). The best fitting model was determined by comparing the R^2^ values. For GEF, the model followed was zero order for all the pH values studied, and the same type of release was determined for PTXL. The release kinetics was different for the release of GEF and PTXL from the dual drug loaded system, but still the overall release model was Zero order. With the Krosmeyer-Peppas model the Fickian release was observed for GEF and PTXL released from GEF-CSCaCO_3_NP and PTXL-CSCaCO_3_NP in the pH of 7.4, 6.5, and 5.6, respectively. Non-Fickian release was observed for GEF and PTXL from the GEF-PTXL-CSCaCO_3_NP in all the pH of 7.4, 6.5, and 5.6 proving that the concentration of the drug was not affecting the release of the drugs from the dual drug delivery system.

### 3.8. Dissolution Analysis of CSCaCO_3_NP

The dissolution profile of CSCaCO_3_NP (60 mg) in phosphate buffered saline with 0.2% (*v*/*v*) Tween 80 and in simulated plasma conditions for a period of 24 h in pH of 7.4, 6.5 and 5.6 were determined. Calcium was released at a higher concentration at the pH of 5.6 and remains the highest, followed by the concentration detected at the pH of 6.5 and 7.4, whereas the dissolution profiles remain highest at the pH of 5.6 in the simulated plasma condition followed by the concentration detected at the pH of 6.5 and 7.4 (Figure 6f).

### 3.9. Alkalinization Profiles of CSCaCO_3_NP

The alkalinization capacity of CSCaCO_3_NP in plasma simulating conditions was determined for 24 h (Figure 7a) and in phosphate buffered saline with 0.2% (*v*/*v*) Tween 80 (Figure 7b) at pH of 7.4, 6.5, and 5.6 with concentrations of 5.00, 1.66 and 0.83 mg/mL. The neutralization capacity of CSCaCO_3_NP in plasma simulation conditions at the end of 24 h was highest for the 5 mg/mL at the starting pH of 5.6, reaching up to the pH of 6.6. Less appreciable neutralization was observed at the same concentration at the starting pH of 7.4, reaching up to 7.6. The neutralization capacity of CSCaCO_3_NP in phosphate buffered saline with 0.2% (*v*/*v*) Tween 80 at the end of 24 h was highest for the 5 mg/mL at the starting pH of 5.6, reaching up to the pH of 8.4. Appreciable neutralization was observed at the same concentration at the starting pH of 7.4, reaching up to the pH of 8.9.

## 4. Discussion

Calcium carbonate nanoparticles have been synthesized by both top-down [35,45,49] and bottom-up methods [50,51]. Each method has its advantages and disadvantages. A critical consideration is that the formulation process must be robust enough to ensure high reproducibility and be streamlined to allow for the ease of scale-up production. Care must be taken that the physicochemical properties of synthesized nanoparticles be preserved throughout the formulation process. Some of the popular method for synthesizing calcium carbonate nanoparticles viz. precipitation [43], sol-gel method [52,53], and layer by layer deposition [51]. As explained by the name, the top-down method is where larger particles (blood cockle shells) are broken down into smaller nanometer-sized particles. This study involves the usage of both the chemical and the mechanical processes to synthesize the CSCaCO_3_NP. Researchers have used milling technology to synthesize poorly water-soluble compounds [54]. In the initial stages of the nanoparticle synthesis, Tween 80 was used as the nonionic surfactant to keep the nanoparticles from growing and aggregate during the vigorous stirring step. Kamba et al. [47] and Isa et al. [40] have used Tween 80 to synthesize CSCaCO_3_NP by the high-pressure homogenization method. In contrast, other researchers have used BS-12 to synthesize the CSCaCO_3_NP from cockle shells [39,49].

Milling is an acceptable fine particle production method where mechanical energy is applied to the solid material to break the bonding between the atoms or molecules [55]. The mechanical dry milling step becomes essential to break down the particles further. The usage of ceramic balls and the balls constant hitting with the container while milling helps break the particles further. Besides, this method is more effective and less expensive than a high-pressure homogenizer technique [35]. Loading the hydrophobic drug like PTXL and GEF onto CSCaCO_3_NP involved a simple stirring method and was effective. Since both GEF and PTXL are hydrophobic drugs, DMSO with Tween 80 buffer (0.05%) was used as solvent for drug loading.

The physicochemical characterization: size, PDI, surface chemistry, drug loading and encapsulation, chemical composition and impurities present must be known to determine the stability and the biological effects of the synthesized nanoparticles [55]. GEF has been quantified using UV spectrophotometry [56,57], liquid chromatography-tandem mass spectrophotometry [58], and high-performance liquid chromatography (HPLC)-UV [59]. PTXL has been quantified using UV-Vis spectrophotometry [60], and HPLC [61,62,63]. Methods such as simultaneous equation, Q absorbance [64], difference spectrophotometry, and derivative spectrophotometry [65] are available for simultaneous estimation of various drugs [66]. Current UV-Vis spectrophotometry, though old-fashioned, was chosen because it is facile, sensitive, less time-consuming, and cost-effective. To date, there is no literature reported on simultaneous quantification of GEF and PTXL in a nanoparticulate formulation, which is the novelty here. UV-Vis spectrophotometer was used for the measurement of GEF [67,68,69] and PTXL [70,71,72] in various other nanoparticles for determining the loading content and/or encapsulation efficiency. The λ_max_ of 332.6 ± 0.2 nm and 247.8 ± 0.3 nm was satisfactory for GEF and PTXL and, this λ_max_ was similar than the wavelength used by other researchers [59,60,63,69]. The developed method was validated and the analytical performance criteria for linearity (Table A3), recovery and precision (Table A4 and Table A5) and sensitivity (Table A5) were found to be reliable and meeting the recommendations of the International Conference on Harmonization Q2(R1) guidelines [46].

Drug-loading and drug-entrapment percentages are vital parameters in the synthesis of nanomedicines [73]. In the three formulations GEF1, GEF2 and GEF3, it was found that the loading content was highest when the concentration of nanoparticles was lowest, and it reduced further from 2.8 ± 2.6 (%), 2 ± 1 (%) to 1.5 ± 1.1 (%), respectively when the quantity of nanoparticles was increased. Less drug availability for loading and hence the reduced loading percentage. A similar trend was observed in the three formulations for PTXL, namely, PTXL1, PTXL2 and PTXL3 nanoparticles, where the loading content reduced from 1.4 ± 0.9 (%), 1 ± 0.1 (%) to 0.4 ± 0.1 (%), respectively, when the quantity of nanoparticles was increased. For the dual drug-loaded GEF1-PTXL1, GEF2-PTXL2 and GEF3-PTXL3 group of nanoparticles, a similar trend in the decrease of loading content was observed from 2 ± 0.1/0.9 ± 0.1 (%), 1.1 ± 0.2/0.5 ± 0.1 (%) to 1.1 ± 0.1/0.4 ± 0.1 (%), respectively for GEF/PTXL (Figure 2b).

Interestingly, the loading content of GEF was always higher when compared to the loading content of PTXL, and it is observed when the concentration of drug and CSCaCO_3_NP are kept similar. The loading efficiency of drugs into the nanoparticles is also governed by the surface area available [74] on the nanoparticles. Another critical factor is the water solubility of the drugs employed. The lower loading content of less than 10% is usually observed for inorganic carrier-based nanoparticles [73], concurrent with the current data. Other researchers have obtained lower loading content of 2.4 ± 0.3% to 7.6 ± 0.9% [75] for GEF-loaded PLGA microspheres, 0.3 ± 0.1% for DOX-calcite hybrid crystals [76], and 0.09% to 0.39% for human recombinant parathyroid hormone-loaded CSCaCO_3_NP [38]. Besides the above-stated facts, other points to be considered for lower LC are the physical and electrostatic interaction during the drug loading process. The CSCaCO_3_NP is negatively charged and both GEF and PTXL are negatively charged. This negative charge could contribute to electrostatic repulsion leading to lower loading content. Another factor is the concentration of drugs used during the loading process. On the other hand, higher LC is required so that the amount of carrier nanoparticles is reduced. Since we aim to harvest the alkalinizing property of calcium carbonate and GEF and PTXL’s potential benefits in the same system, the lower loading content is accepted for future research This point also facilitates the prospects of synthesizing CSCaCO_3_NP with higher LC by modifying the synthesis technique of CSCaCO_3_NP, drug loading process, or surface functionalization of nanoparticles.

A fraction of the drug that is incorporated into the nanoparticles relative to the total amount of drug added is defined as entrapment efficiency [77]. The entrapment efficiency for GEF in GEF1, GEF2 and GEF3 nanoparticles was lowest when the concentration of nanoparticles was lower, and it increased further from 47.4 ± 20.2 (%), 52.3 ± 18.1 (%) to 53.4 ± 17.6 (%), respectively when the quantity of CSCaCO_3_NP was increased (Figure 2a). For PTXL in PTXL1, PTXL2 and PTXL3 the entrapment efficiency was 32.1 ± 10.8, 36.8 ± 3.7, and 20.8 ± 1.6, respectively. For the dual drug-loaded GEF1-PTXL1, GEF2-PTXL2 and GEF3-PTXL3, a trend of decrease and then increase of entrapment efficiency was observed from 50 ± 2.1/45.6 ± 0.3 (%), 43 ± 8.9/37.5 ± 5.7 (%) to 45 ± 10.3/43.9 ± 7.2 (%), respectively for GEF and PTXL (Figure 2a).

Out of the three formulations, GEF2 is selected since it has a higher entrapment efficiency 52.3 ± 18.1 (%) and higher loading content 2± 1 (%) when compared with the other groups. Similarly, PTXL2 is suitable with its highest encapsulation efficiency 36.8 ± 6.5 (%) and higher loading content 1 ± 0.1 (%). On the other hand, GEF1-PTXL1 will be suitable since it possesses higher entrapment efficiency (50 ± 2/45.6 ± 0.3%) and the highest loading content (2 ± 0.1/0.9 ± 0.1%) of GEF and PTXL, respectively. The observed trend in the LC (%) and the entrapment efficiency (%) was similar to that data obtained for a few anti-cancer drugs like doxorubicin [33,34] and docetaxel [37] loaded onto the blood cockle shell derived CaCO_3_NP. This group of nanoparticles were used for the characterization studies.

The size of nanoparticles is the most crucial parameter to be considered while designing nanomedicines. The size affects the particles behaviour in vivo, viz. internalization into the cells, deposition, and reticuloendothelial clearance. The shape of the nanoparticles also is an important criterion that will define how the drug will behave In vitro and in vivo. Usually, the abiogenic aragonites are rod or needle-shaped, but the biogenic aragonites are sometimes elliptical shapes, as observed by Falini et al. [78]. Another study on the arrangement of cross lamellar layers of molluscan shells indicates that the aragonites form elongated rod-shaped crystals [40]. Islam et al. [79] and Kamba et al. [47] have synthesized rod-shaped cockle shell-derived aragonite calcium carbonate nanoparticles. Electron microscopic studies reveal that the CSCaCO_3_NP are mostly spherical, which agrees with the findings of other studies involving calcium carbonate nanoparticles [33,49]. Compared to the size of cockle shell-derived CaCO_3_ NP synthesized by Isa et al. (13.94–23.95 nm) [40] and Danmaigoro et al. (24 ± 4.07 nm) [33], the CSCaCO_3_NP synthesized in this study are larger (63.9 ± 22.3 nm). The spherical shape of the plain and drug-loaded CSCaCO_3_NP is in coherency with other researchers.

The hydrodynamic diameter of synthesized nanoparticles was greater than the doxorubicin-loaded CSCaCO_3_NP (185.6 ± 2.4) obtained by Danmaigoro et al. [33] but smaller than oxytetracycline-loaded CSCaCO_3_NP (276 ± 6.3) [80]. The negative ζ-potential is concurrent with the findings of other researchers [33,80]. According to Bhattacharjee et al., the Zeta potential measures the surface potential of the nanoparticles and not the charge or the charge density. Hence the value of the zeta potential is vital when compared to the negative/positive value associated with the potential. The majority of naturally occurring surfaces and molecules exhibit negative charge, and hence no wonder calcium carbonate exhibits a negative zeta potential. Various other parameters play an important role in the determination of the zeta potential like pH of the solvent, ionic strength, and the concentration of nanoparticles [81]. In contrast, positively charged calcium carbonate nanoparticles were synthesized by Lee et al. where the source of calcium carbonate was ground seashells. The same study gives the striking data of the size measured using SEM and DLS to be ~30 nm and 2187 nm [82], where the size measured with SEM is about 72 times lesser than the size obtained by DLS.

When we examined the hydrodynamic diameters, and the size measured using electron microscopy from past research with cockle shells, we could calculate up to a 17–31% increase in the size measured using DLS when compared to the size measured by electron microscopy [33,34,80]. When we compare the data of the size of the particles from the electron micrographs and hydrodynamic diameter from DLS, it is evident that the latter value is relatively higher, the reason for this discrepancy have been already explained [83]. Hydrodynamic diameter data is based on the intensity of light scattering fluctuations by the particles. In contrast, in TEM, the scattered electrons’ size is measured from the particles under a vacuum. For SEM, the size’s measurement is from the electrons emitted from the nanoparticles under the vacuum. The metal coating carried out during SEM was found to introduce the error rate up to 14 nm [84]. The TEM and SEM gives us the two-dimensional projection image of the nanoparticles. The broad size distribution observed in the hydrodynamic diameter of CSCaCO_3_NP and drug-loaded CSCaCO_3_NP can be attributed to various factors such as the aggregation or opsonization of particles in the liquid medium (here, deionized water) [55,85]. GEF and PTXL in PBS with 1% Tween 80 was found to be negatively charged. The PDI is the ratio between mean particle size and standard deviation, and it reflects the homogeneity of the size distribution [82]. Though the PDI obtained for the synthesized particles was ~0.3, we cannot rule out the possibility that there could be a polydisperse population since we follow the top-down method of nanoparticle synthesis.

The EDX data of the CSCaCO_3_NP, revealed that the Ca/C/O was ~1:1:3, indicative of pure calcium carbonate. Hazmi et al. [21] have reported 1.3% of Mg, Na, P, K, and other elements (Fe, Cu, Ni, B, Zn, and Si) in the blood cockle shells of western peninsular Malaysia. Another study reported detection of Al and S peaks in addition to the Ca/C/O [86]. When we analyzed the EDX data of drug-loaded nanoparticles, the ratio of Ca/C/O was not ~1:1:3, and the elemental percentage of calcium was less than that of the CSCaCO_3_NP which suggests the presence of other elements which could be due to the surface functionalization with the drugs. It is interesting to note that when GEF with the molecular formula of C_22_H_24_ClNFN_4_O_3_ [87] was analyzed, oxygen (10.85%), carbon (68.33%), nitrogen (14.02%), fluorine (4.05%), and chlorine (2.28%) were obtained. For PTXL, with the molecular formula C_47_H_51_NO_14_ [88], only oxygen (16.37%) and carbon (80.63%) were inferred from the spectra, where the N peak could not be observed, even after repeated measurements at different sites.

This XRD result of synthesized nanoparticles show that they are aragonite crystals. The results are in agreement with other researchers where various other drugs like vancomycin [39], doxorubicin [33], thymoquinone, and doxorubicin [89] were loaded onto the CSCaCO_3_NP and the aragonite phase remained unaltered. Other researchers have found that the crystallites are arranged in a cross lamellar pattern, as observed in other Mollusca and bivalves belonging to the same Arcidae family [90,91,92]. Furthermore, this is the cockle shells’ microstructure, broken down by the mechanical milling to a nanometer size. From the paper published by Bragg in 1924, one could say that the aragonite and calcite share the same chemical formula and similar diffraction patterns but vary significantly in their crystalline arrangement [93]. A study in 2019 by Sampath et al. [92] proved the presence of a calcite layer and an aragonite morphology in *Anadara inaequivalvis* shells. Two studies by Yang et al. [94] and Chong et al. [95] revealed that there occurs a strong peak at 2θ of 29.4, indicative of calcite, and this peak disappeared when the calcium carbonate became aragonite. In a paper published by Hussein et al. [53] where the cockle shells were used to synthesize nanoparticles using a sol-gel method, this resulted in a calcite phase, characterized by the peak of 2θ at 29.5, was the confirmatory sign of calcite. In the current study, we found no diffraction around that region in the diffractogram of the nanoparticles synthesized, which indicates that these nanoparticles are purely aragonite. The crystallite size of the synthesized nanoparticles ranged from 30.1 nm to 32.8 nm. In a 2020 study, calcium carbonate nanoparticles of crystallite size of 61.4 nm were synthesized from cockle shells as the source of calcite calcium carbonate nanoparticles [53]. Another study by Lee et al. [82] shows that calcite calcium carbonate nanoparticles were ~30 nm, which is in the same range as the crystallite size obtained in this study.

To investigate the interaction between the CSCaCO_3_NP and the drugs, FTIR was carried out. Spectral analysis provides us with the position, shape, and relative intensities of the vibrational bands. FTIR spectra result from the absorption of electromagnetic radiations at frequencies that correlate to the vibration of a specific set of chemical bonds form within a molecule [96]. It is interesting to note that the chemical bonds formed between the nanomaterials and the drugs are usually hard to interpret following the conventional database of known vibrational frequencies, so it is vital to determine the relationship between the drug and the nanoparticles. The calcium carbonate nanoparticles we have synthesized exhibit the conventional peaks of the carbonate ion (CO_3_^2−^). The carbonate ions show four significant vibration modes (ν_1_ to ν_4_) between 600 and 1500 cm^−1^ [97,98] (ν-stretching). The reported vibrational frequencies of Ca(CO_3_) belonging to aragonite crystalline symmetry is as follows: 1080 (ν_1_), 866 (ν_2_), 1504, 1492 (ν_3_) and 711, 706 (ν_4_) (Table 5) [99]. When we look at the structure and bonds present in a calcium carbonate molecule, there is the ionic bond between the Ca^2+^ and the CO_3_^2−^ polyatomic ion. Also, there exists covalent bonding between the non-metallic atoms of carbon and oxygen (CO_3_^2−^). Though the cationic coordination in aragonite is nine, the ionic cations’ electrostatic forces have a negligible effect on the more strongly bound anions’ internal vibrations. The anion-anion forces play a significant role in bringing about changes in the vibrational frequencies rather than anion-cation forces [100]. The salient point is that using the IR spectra can differentiate between the calcite and the aragonite, as calcite sometimes lacks the ν_1_ mode of vibrations and the characteristic vibrational band assignment for aragonite: ν_1_ at 1083 cm^−1^, and ν_4_ is a doublet [97] at 714 and 700 cm^−1^ [99].The presence of a peak at 858 cm^−1^ in the spectra of non-biogenic calcium carbonate is a tell-tale sign of the aragonite phase [101].The ν_2_ and the doublet peak confirm crystalline aragonite instead of an amorphous state [102]. The absorption peaks at 710 and 872 cm^−1^ are always associated with the calcite phase [103]. The result of the PWRD for the CSCaCO_3_NP in the current work correspond to the aragonite phase. A 2002 study on the marine mussel *Arca brunesi* mentioned that the band at 1082 cm^−1^ is observed only in the aragonite crystalline matrix [98], while on the other hand, the absence of a band around this region has been attributed to calcite. In a study where the researchers synthesized non-biogenic calcium carbonate, a peak at 1089 cm^−1^ (quite close to 1082 cm^−1^) was observed in the calcite spectra [95].

The spectral absorption peaks of GEF-CSCaCO_3_NP demonstrated less pronounced 3412 cm^−1^ (N-H/O-H stretch), 1024 (C-F stretch) and 953 cm^−1^ (alkane/aromatic deformation of C-H group) cm^−1^ peaks as compared to GEF-PTXL-CSCaCO_3_NP. The spectral absorption peaks of PTXL-CSCaCO_3_NP demonstrated a slightly more pronounced ν_2_ band, 953 cm^−1^ (cyclohexane/alkane/C-H in-plane deformation) and a weak 1022 cm^−1^ (C-N stretching) band. The spectral absorption peaks of GEF-PTXL-CSCaCO_3_NP demonstrated new vibrational bands at 953 cm^−1^ (cyclohexane/alkane), broader 1024 cm^−1^ (C-F/C-N stretch), very weak 2918 cm^−1^ (CH_3_/C-H/alkanes anti-symmetric stretch), and weak 3435 cm^−1^ (N-H/O-H stretch) (Figure 5) bands [96,104,105].The largest and strongest band exhibited by CSCaCO_3_NP at 1445 cm^−1^ remained unaltered in the spectra of GEF-PTXL-CSCaCO_3_NP, indicating that the alkyl group is unaffected. All these changes indicate that the drugs have successfully bound to the nanoparticles.

Nitrogen adsorption and desorption experiments are usually used to determine the pore characteristics and the nature of adsorbent/adsorbate interactions. When the experiment is carried out at a constant temperature, the amount adsorbed is recorded as the adsorptive pressure or concentration function. The relation between the adsorbed amount of gas and the gas’ equilibrium pressure at a constant temperature is called the adsorption isotherm [106]. The isotherm quality is the first characteristic to be noted when analyzing adsorption data. The isotherms provide valuable data about the pore characteristics and the nature of adsorbent/adsorbate interactions. All the isotherms obtained from the synthesized nanoparticles were Type IV and was classified based on the BET classification system and the International Union of Pure and Applied Chemistry (IUPAC) manual [107]. The BET method has its weakness in the theoretical assumptions. Nevertheless, it is one of the most widely used methods for evaluating porous materials’ surface area and porosity. This type of isotherm is characterized by the “hysteresis loop,” where capillary condensation occurs, with an initial loop formed by the mono-multi layer adsorption, a 2nd loop by the desorption of gases. This type of isotherm indicates that the nanoparticles are mesoporous [108,109]. All the isotherms produced by the nanoparticles show hysteresis at the high-pressure region, indicating the isotherm reliability for data validity. The hysteresis loop was very narrow, with the adsorption and desorption branches being nearly vertical and parallel above 0.8 relative pressure (Figure 3). The hysteresis loop is a mixture of H1 and H3. The two branches are almost vertical and nearly parallel to each other, but they lack the plateau region towards higher relative pressure. We designated both H3 and H1 as H3 type of hysteresis does not have the plateau and resembles the hysteresis loop of the isotherms obtained. Singh and Williams’s paper indicates that H3 cannot be a Type IV due to plateau’s presence in the latter [110]. Type IV was the appropriate type since no other isotherm type coincides with the isotherms obtained in this study. As per the IUPAC recommendation, the materials containing pore widths between 2 nm and 50 nm are classified as mesopores. The BET area of the type IV isotherms is considered “true probe accessible surface area” [109]. Though we have designated the isotherms to be type IV, they lack the plateau towards the higher P/P_0_.

The nanoparticles have a higher surface area when compared to the size, this being the reason why nanoparticles are a perfect drug delivery system [111]. There are differences in the surface areas and pore width of calcium carbonate nanoparticles, as shown in Table 6. It could be due to the presence of various elemental and polar surface functional groups [106] in drug-loaded CSCaCO_3_NP. The classical range for application of BET for type IV isotherm is p/p^0^ of 0.05–0.3. It is apparent that the surface area and pore width obtained by researchers who used cockle shells [33,37] as the primary source of calcium carbonate remains in a similar range as the result from this study and do not vary significantly. It is interesting to observe that the amorphous calcium carbonate nanoparticles synthesized by Sun et al. [112] have the highest surface area regarding calcium carbonate nanoparticles. However, these researchers have synthesized amorphous calcium carbonate nanoparticles, which cannot be compared to the crystalline counterparts.

In vitro drug release studies of GEF and PTXL were carried out in PBS with 0.2% (*v*/*v*) Tween 80 at pH 6.5 [32], 5.6 [76,113], and 7.4 at 37 °C by the dialysis method [48]. The drug release data suggests that: (1) the drug will be released from the nanoparticles at a higher concentration inside the cancer cell when compared to the release in the normal physiological pH, and (2) sustained release kinetics of the drugs for a period of 100 h could be achieved, as the physical structure of the calcium carbonate nanoparticles and the pores facilitate this release behaviour. Therefore, we believe that the drug released from the GEF-PTXL-CSCaCO_3_NP will facilitate synergistic release of GEF and PTXL in a synergistic manner to achieve the speculated synergistic effects. The lower release data of this study is in concurrence with the release data obtained for paclitaxel in thiol-terminated PEG-paclitaxel-conjugated gold nanoparticles where only 10% release was obtained for a period of 120 h [114]. This is similar to the release observed in the study of Wu et al. [42] where calcium carbonate nanoparticles were loaded with PTXL and doxorubicin hydrochloride. It is thought that the difference in the release is due to the hydrophilic nature of calcium carbonate and the loaded drugs, so in comparison with the pure drugs, permeation of liquid molecules into the nanoparticles and afterwards diffusion of the drugs to the external release medium becomes more difficult. No burst release of drugs was observed due to the chemical nature of GEF and PTXL being hydrophobic.

As per the article by Mircioiu et al. [115] the modeling and prediction of the release kinetics represent higher complexity and involves tremendous understanding of the physicochemical, physiological, and mathematical aspects of the release kinetics. The in vitro dissolution and release studies are carried out to examine the drug release model of the newly synthesized nano formulations. The effect of dialysis membrane in underestimating the release of drug has been found. The reason why the release kinetics determined using the dialysis method can be misleading compared to sustained release of drug from nanocarriers is that the drug though is dissolved into the dialysis compartment, but the apparent drug release remains slow due to the slow drug permeation from the dialysis compartment to the external solvent compartment [116]. According to Ma-Ham et al. [117] the dialysis rate of the drugs is determined by the molecular size of the drugs, pore size of the dialysis membrane, solubility of the drugs. The flow rate is faster if the drugs are smaller, hydrophilic, and the dialysis membrane has a larger pore size. In the current study, the dialysis membrane of 14,000 MWCO was used, which has adequate pore size for the entry of the solubilized drugs GEF (446.9) and PTXL (853.9) [88] with the molecular weight much lower than the pores. However, we noticed that the release of the drug was slow and consistent.

The alkalinization or neutralization ability of CSCaCO_3_NP studied in plasma simulating conditions prove that the highest alkalization occurs at the lower pH of 5.6 when compared to the pH of 7.4, whereas in phosphate buffered saline with 0.2% (*v*/*v*) Tween 80 the alkalinization ability was found to be higher than the physiological pH of 7.4. Som et al. [19] have suggested that CaCO_3_ nanoparticles increase the pH to about 7.4, which is the physiological pH and did not bring about metabolic alkalosis in the surrounding healthy tissue. The buffering capacity observed in the plasma simulating condition at various pH confirms the speculation that the CSCaCO_3_NP will help in alkalizing the tumor microenvironment and hence reduce the incidence of metastasis and tumor growth. The dissolution profile of CSCaCO_3_NP (60 mg) in phosphate buffered saline with 0.2% (*v*/*v*) Tween 80 and in simulated plasma conditions revealed that the dissolution was the highest in the pH of 5.6 and in the plasma simulating condition, when compared to the concentration detected in the 0.2% Tween 80 PBS of the same pH. Calcium was released at a higher concentration in the pH of 5.6 and remains the highest, followed by the concentration detected in 6.5 and 7.4. In a study by Pingitore et al. [118] calcium carbonate in the form of calcite and aragonite were tested for their solubility in various solvents including deionized water. They found that coarsely grained aragonite and calcite phases of non-biogenic calcium carbonate dissolved in deionized water and 15–20 ppm of calcium ions was released in 200 h from 0.1 g of sample of calcium carbonate. The dissolution of CaCO_3_ follows the basic rate equation with the formation of calcium ions and carbonate ions. In the previous studies, it has been shown that the CaCO_3_ nanoparticles consistently are soluble in mildly acidic media and served as a buffering agent. The cellular influences of calcium carbonate will be from the calcium ion released and the drug released when they are dissolved in the extracellular or intracellular milieu. Other studies indicate that the calcium carbonate nanoparticles enter the cell by energy-dependent endocytosis [119]. Since calcium functions as an intracellular signal transducer, the release of calcium from the CSCaCO_3_NP could disrupt cellular signaling and hence resulting in cellular changes: induction of ROS, increase in CHOP, caspase-3 activity, and ER stress. Calcium is deemed to be least toxic among its many metal allies like Zn^2+^, Ni^2+^, and Cu^2+^ as calcium s an essential element for humans and there already exists a regulation system for the metabolism of calcium [120]. The synthesized CSCaCO_3_NP have attractive characteristics such as higher solubility in mildly acidic pH, drug loading capacity, aragonite crystalline phase which will make scientists choose CSCaCO_3_NP when compared to other metal-based nanoparticles for designing drug delivery systems. Moving on to the drugs, as GEF and PTXL loaded onto CSCaCO_3_NP will be released in the tumor site due to the pH-dependent solubility of calcium carbonate nanoparticles. In a study where doxorubicin hydrochloride and PTXL were co-encapsulated in an inorganic/organic hybrid alginate/CaCO_3_ nanoparticles, dual drug-loaded nanoparticles exhibited significant enhanced cell uptake and nuclear localization compared with the single drug-loaded nanoparticles [42]. Hence, embracing the possibility of synthesizing calcium carbonate nanoparticles with their distinguishing characteristic features and loading them with GEF and PTXL would be the first step in formulating nanomedicines, as a targeted therapy against breast cancer.

## 5. Conclusions

We have developed a facile, scalable method without using harsh chemicals to produce CSCaCO_3_NP from the waste shells of blood cockles. Dry ball milling effectively results in nanoparticles of suitable size, morphology, and crystalline phase. We have demonstrated that we could load GEF and PTXL resulting in GEF-CSCaCO_3_NP and PTXL-CSCaCO_3_NP. For the first time, we have simultaneously loaded GEF and PTXL to CSCaCO_3_NP produce dual drug-loaded GEF-PTXL-CSCaCO_3_NP. We have successfully characterized the synthesized nanoparticles, which reveals that they are spherical, porous, and possess a large surface area compared to the particle size. The FTIR shows that the drugs are bound to CSCaCO_3_NP. The drug loaded CSCaCO_3_NP have prolonged release and the CSCaCO_3_NP show highest solubility in moderately acidic pH solution with alkalization properties in the plasma simulating condition. Though several drug-loaded CSCaCO_3_NP have been synthesized, dual-drug loaded GEF-PTXL-CSCaCO_3_NP should have a place in cancer research and treatment due to the potential they possess.

## Figures and Tables

**Figure 1 nanomaterials-11-01988-f001:**
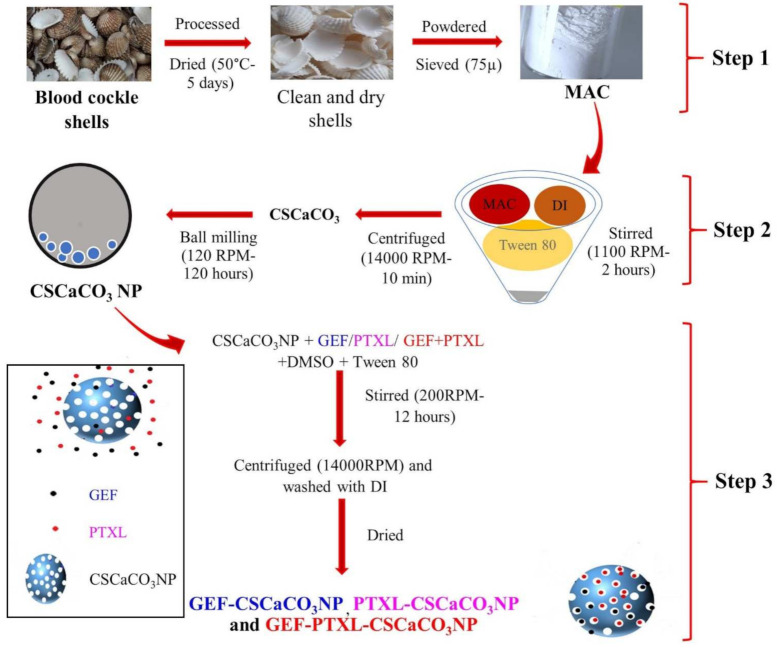
Schematic diagram showing the step-by-step synthesis of Blood cockle shell derived CaCO_3_ nanoparticles. Step 1: Transformation of blood cockle shells into MAC, Step 2: Synthesis of CSCaCO_3_NP from MAC, and Step 3: Synthesis of Mono drug-loaded GEF-CSCaCO_3_NP, PTXL-CSCaCO_3_NP, and dual drug-loaded GEF-PTXL-CSCaCO_3_NP. MAC, micron-size aragonite calcium carbonate particles; DI, Deionized water; CSCaCO_3_NP, Blood cockle shell derived CaCO_3_ nanoparticles; GEF, Gefitinib; PTXL, Paclitaxel; GEF-CSCaCO_3_NP, GEF loaded Blood cockle shell derived CaCO_3_ nanoparticles; PTXL-CSCaCO_3_NP, PTXL loaded Blood cockle shell derived CaCO_3_ nanoparticles and GEF-PTXL-CSCaCO_3_NP, GEF and PTXL loaded Blood cockle shell derived CaCO_3_ nanoparticles.

**Figure 2 nanomaterials-11-01988-f002:**
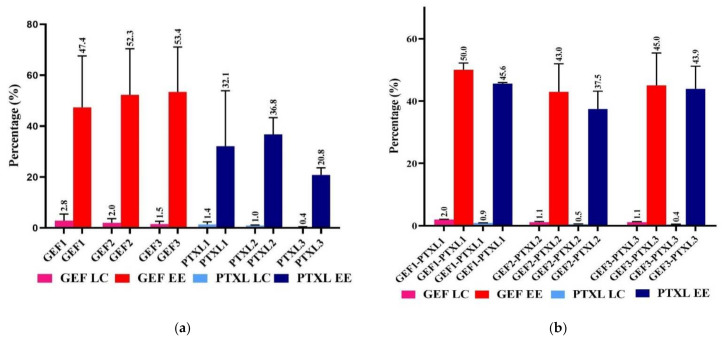
Loading Content (LC) and Entrapment efficiency (EE): (**a**) Mono GEF and PTXL in three individual groups of CSCaCO_3_ nanoparticle formulations; (**b**) Dual GEF+PTXL in three groups of CSCaCO_3_ nanoparticle formulations. The error bar is indicated for the corresponding columns, and the mean data are shown above the error bar. LC, loading content; EE, entrapment efficiency; CSCaCO_3_NP, Blood cockle shell derived CaCO_3_ nanoparticles; GEF, Gefitinib; PTXL, Paclitaxel; GEF 1-3, groups of GEF loaded CSCaCO_3_NP; PTXL 1-3, groups of PTXL loaded CSCaCO_3_NP; GEF 1-3, PTXL 1-3, and GEF1-3-PTXL1-3 are groups of GEF loaded CSCaCO_3_P, PTXL loaded CSCaCO_3_NP, and GEF-PTXL loaded CSCaCO_3_NP.

**Figure 3 nanomaterials-11-01988-f003:**
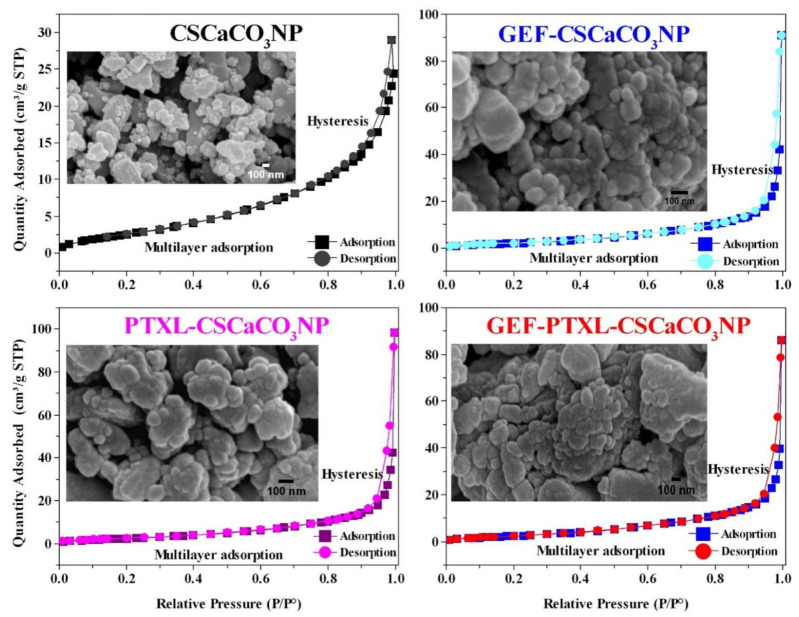
FESEM representative images of CSCaCO_3_NP, GEF-CSCaCO_3_NP, PTXL-CSCaCO_3_NP, and GEF-PTXL-CSCaCO_3_NP (Scale bar: 100nm) and the corresponding nitrogen adsorption desorption isotherms with its characteristic hysteresis. Plotted with Relative pressure (P/P^0^) on the *x*-axis and Amount of Nitrogen adsorbed (cm^3^/g) on the *y*-axis.

**Figure 4 nanomaterials-11-01988-f004:**
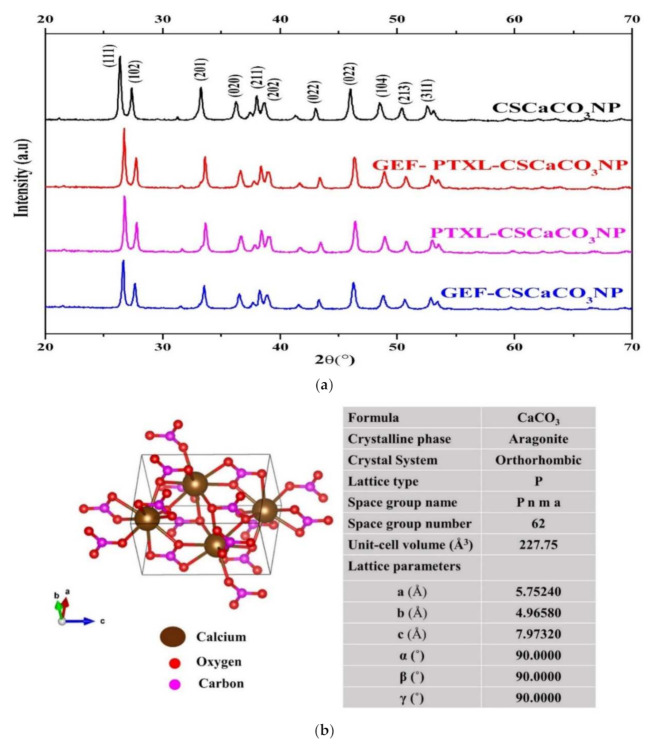
Powder X-ray diffractograms (XRD): (**a**) XRD patterns of CSCaCO_3_NP, GEF-PTXL-CSCaCO_3_NP, GEF-CSCaCO_3_NP, and PTXL-CSCaCO_3_NP. Labelled are the miller indices (h, k, l) planes indicating the peaks of the aragonite crystalline phase; (**b**) Orthorhombic crystalline morphology of the synthesized aragonite CSCaCO_3_NP along with its lattice parameters.

**Figure 5 nanomaterials-11-01988-f005:**
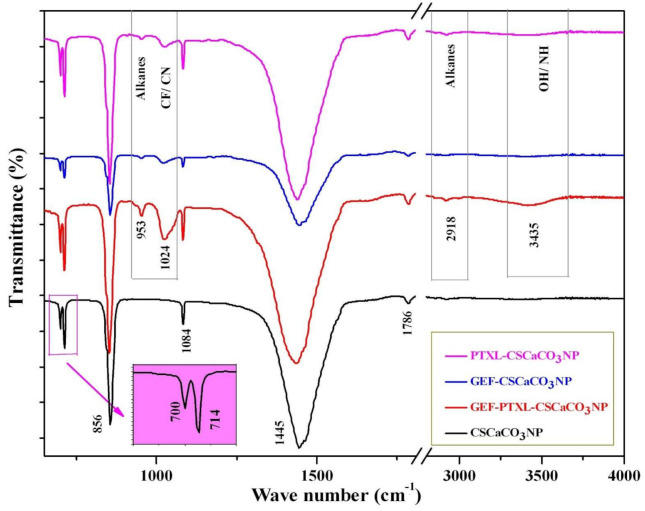
Fourier-transform infrared spectroscopy FTIR patterns of CSCaCO_3_NP (*y*-axis: 15–100%, corresponding to the lowest point of curve and the highest), GEF-PTXL-CSCaCO_3_NP (*y*-axis: 6–100%, corresponding to the lowest point of curve and the highest), GEF-CSCaCO_3_NP (*y*-axis: 10–100%, corresponding to the lowest point of curve and the highest), and PTXL-CSCaCO_3_NP (*y*-axis: 5–100%, corresponding to the lowest point of curve and the highest). The insert shows the doublet ν_4_ the peaks (Pink shaded box), the newly formed peaks are indicated with the grey box (break in *x*-axis: 1810–2800 cm^−1^).

**Figure 6 nanomaterials-11-01988-f006:**
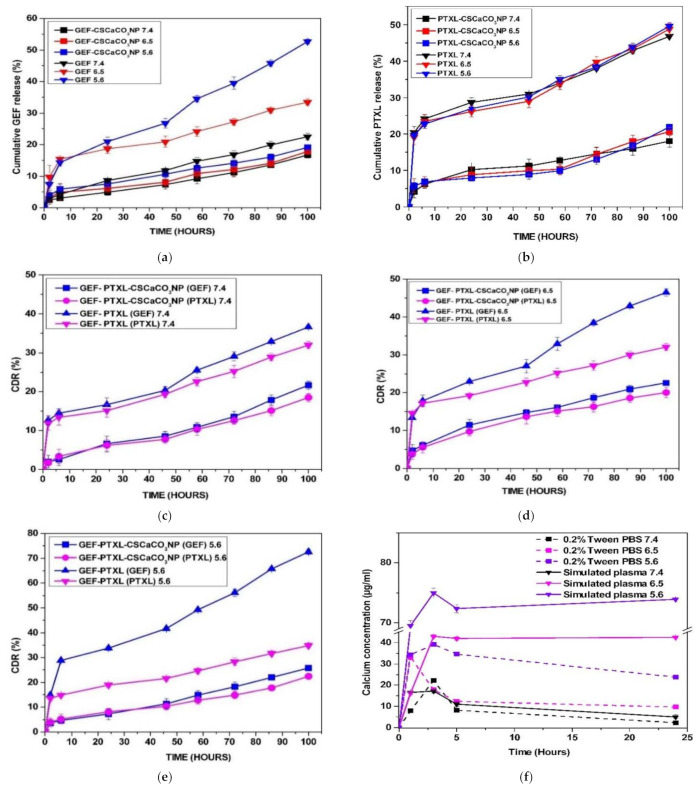
In vitro release of: (**a**) GEF from GEF-CSCaCO_3_NP in PBS with 0.2% (*v*/*v*) Tween 80 at pH of 5.6, 6.5, and 7.4; (**b**) PTXL from PTXL-CSCaCO_3_NP PBS with 0.2% (*v*/*v*) Tween 80 at pH 6.5, 5.6, and 7.4; (**c**) GEF and PTXL from GEF-PTXL-CaCO_3_NP in PBS with 0.2% (*v*/*v*) Tween 80 at pH of 7.4; (**d**) GEF and PTXL from GEF-PTXL-CaCO_3_NP in PBS with 0.2% (*v*/*v*) Tween 80 at pH of 6.5; (**e**) GEF and PTXL from GEF-PTXL-CaCO_3_NP in PBS with 0.2% (*v*/*v*) Tween 80 at pH of 5.6; and (**f**) Calcium concentrations released from CSCaCO_3_NP in PBS with 0.2% (*v*/*v*) Tween 80 and simulated plasma conditions at pH of 5.6, 6.5, and 7.4 (Break in *y*-axis: 45–69).

**Figure 7 nanomaterials-11-01988-f007:**
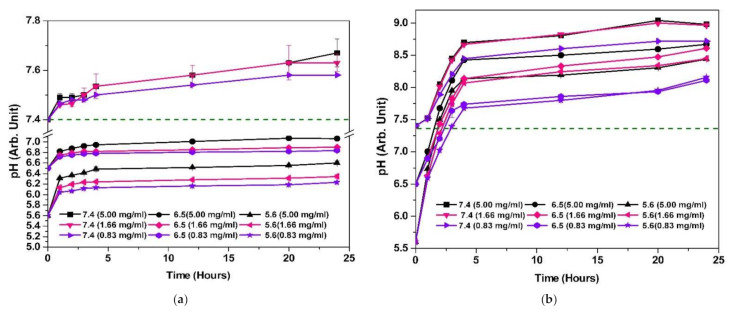
Alkalinization profiles of CSCaCO_3_NP: (**a**) In simulated plasma fluid at pH of 7.4, 6.5 and 5.6 and concentrations of 5.00 mg/mL, 1.66 mg/mL and 0.83 mg/mL (Break in *y*-axis: 7.12–7.35) and (**b**) In phosphate buffered saline with 0.2% (*v*/*v*) Tween 80 at pH of 7.4, 6.5 and 5.6 and concentrations of 5.00 mg/mL, 1.66 mg/mL and 0.83 mg/mL. Dotted line indicating the pH of 7.4 in both the figures.

**Table 1 nanomaterials-11-01988-t001:** Experimental design for synthesizing drug-loaded CSCaCO_3_NP.

Sample	Drug	Drug (µg)	Solvent (mL) ^1^	CSCaCO_3_NP (µg)
GEF1-CSCaCO_3_NP	GEF	400	3	10,000
GEF2-CSCaCO_3_NP				15,000
GEF3-CSCaCO_3_NP				20,000
PTXL1-CSCaCO_3_NP	PTXL	400	3	10,000
PTXL2-CSCaCO_3_NP				15,000
PTXL3-CSCaCO_3_NP				20,000
GEF1-PTXL1-CSCaCO_3_NP	GEF+PTXL	400 + 200	3	10,000
GEF2-PTXL2-CSCaCO_3_NP				15,000
GEF3-PTXL3-CSCaCO_3_NP				20,000

^1^ All the samples were maintained at the same temperature, and the drug loading was executed following a simple stirring process; the solvent utilized was DMSO+0.05% Tween80 buffer (1:1 ratio). CSCaCO_3_NP, Blood cockle shell derived CaCO_3_ nanoparticles; GEF, Gefitinib; PTXL, Paclitaxel; GEF-CSCaCO_3_NP, GEF loaded Blood cockle shell derived CaCO_3_ nanoparticles; PTXL-CSCaCO_3_NP, PTXL loaded Blood cockle shell derived CaCO_3_ nanoparticles and GEF-PTXL-CSCaCO_3_NP, GEF and PTXL loaded Blood cockle shell derived CaCO_3_ nanoparticles.

**Table 2 nanomaterials-11-01988-t002:** Elemental composition of CSCaCO_3_NP, GEF-CSCaCO_3_NP, PTXL-CSCaCO_3_NP, GEF-PTXL-CSCaCO_3_NP, GEF and PTXL computed using energy dispersion X-ray (EDX) spectroscopy.

Samples ^1^	Elemental Composition (%)						
	C	O	Na	Ca	N	F	Cl
CSCaCO_3_NP	22.44	50.09	0.21	27.26	ND	ND	ND
GEF-CSCaCO_3_NP	21.23	63.11	0.28	15.37	ND	ND	ND
PTXL-CSCaCO_3_NP	17.53	61.41	0.30	20.76	ND	ND	ND
GEF-PTXL-CSCaCO_3_NP	18.53	61.65	0.30	19.53	ND	ND	ND
GEF	68.33	10.85	ND	ND	14.02	4.53	2.28
PTXL	83.40	16.60	ND	ND	ND	ND	ND

^1^ CSCaCO_3_NP, Blood cockle shell derived CaCO_3_ nanoparticles; GEF, gefitinib; PTXL, paclitaxel; GEF-CSCaCO_3_NP, GEF loaded blood cockle shell-derived CaCO_3_ nanoparticles; PTXL-CSCaCO_3_NP, PTXL loaded blood cockle shell-derived CaCO_3_ nanoparticles; GEF-PTXL-CSCaCO_3_NP, GEF and PTXL loaded blood cockle shell-derived CaCO_3_ nanoparticles; C, carbon; O, oxygen; Na, sodium; Ca, calcium; N, nitrogen; F, fluorine; Cl, chlorine, and ND, not detected.

**Table 3 nanomaterials-11-01988-t003:** DLS results showing the apparent hydrodynamic diameter, polydispersive Index (PDI) and zeta potential.

Sample ^1^	Hydrodynamic Diameter (nm)	PDI	ζ-Potential (mV)
CSCaCO_3_NP	179 ± 11	0.3 ± 0.1	−17.0 ± 1.2
GEF-CSCaCO_3_NP	188 ± 5	0.3 ± 0.1	−21.5 ± 1.3
PTXL-CSCaCO_3_NP	268 ± 20	0.3 ± 0.1	−4.9 ± 3.4
GEF-PTXL-CSCaCO_3_NP	274 ± 23	0.3 ± 0.1	−10.3 ± 1.7
GEF	149 ± 27	0.2 ± 0.1	−4.3 ± 1.0
PTXL	12 ± 1	0.2 ± 0.1	−9.5 ± 2.4

^1^ CSCaCO_3_NP, GEF-CSCaCO_3_NP, PTXL-CSCaCO_3_NP and GEF-PTXL-CSCaCO_3_NP were measured in deionized water. GEF and PTXL were measured in PBS with 1%Tween 80 pH 7.4. The data are expressed as mean ± SD. CSCaCO_3_NP, blood cockle shell-derived CaCO_3_ nanoparticles; GEF, gefitinib; PTXL, paclitaxel; GEF-CSCaCO_3_NP, GEF loaded blood cockle shell-derived CaCO_3_ nanoparticles; PTXL-CSCaCO_3_NP, PTXL loaded blood cockle shell-derived CaCO_3_ nanoparticles; GEF-PTXL-CSCaCO_3_NP, GEF and PTXL loaded blood cockle shell-derived CaCO_3_ nanoparticles.

**Table 4 nanomaterials-11-01988-t004:** Correlation coefficients R^2^ for GEF, PTXL, and GEF-PTXL derived from various kinetics models and the n value for the Krosmeyer-Peppas model in PBS with 0.2% (*v*/*v*) Tween 80 at pH of 5.6, 6.5, and 7.4.

Formulations	Drug	pH	Kinetics Model				
			Zero Order R^2^	First Order R^2^	Krosmeyer-Peppas		Higuchi R^2^
					R^2^	n	
GEF-CSCaCO_3_NP	GEF	7.4	0.9933	0.9904	0.9505	0.42 ^#^	0.9803
		6.5	0.9816	0.9802	0.9360	0.33 ^#^	0.968
		5.6	0.9905	0.9886	0.9616	0.33 ^#^	0.9894
PTXL-CSCaCO_3_NP	PTXL	7.4	0.9766	0.9576	0.9662	0.35 ^#^	0.9695
		6.5	0.9131	0.9012	0.8880	0.21 ^#^	0.9063
		5.6	0.9122	0.8916	0.8527	0.19 ^#^	0.8741
GEF-PTXL-CSCaCO_3_NP	GEF	7.4	0.9761	0.9698	0.9727	0.54 *	0.9680
		6.5	0.9813	0.9776	0.9802	0.50 *	0.9758
		5.6	0.9894	0.9832	0.9257	0.51 *	0.9849
	PTXL	7.4	0.9846	0.9829	0.9734	0.56 *	0.9799
		6.5	0.9878	0.9739	0.9860	0.50 *	0.9722
		5.6	0.9913	0.9891	0.9522	0.59 *	0.9904

^#^ Fickian diffusion, * Non-Fickian diffusion.

**Table 5 nanomaterials-11-01988-t005:** FTIR vibrational assignments of blood cockle shell derived CaCO_3_ nanoparticles.

Normal Vibrational Assignments		Position (cm^−1^)				
		Current Study	^1^	^2^	^3^	^4^
ν_1_	Symmetric C-O stretching	1084	1072.29	1077	NA	~1082
ν_2_	CO_3_ out-of-plane deformation mode	856	854.77	851	855	~855
ν_3_	Asymmetric C-O stretching mode	1445	1452	1444.05	1455.16	~1455
ν_4_	OCO (in-plane deformation) bending	714 700	707.58	707	708	~709
ν_1_ + ν_4_		1786	NA	NA	NA	1786

^1^ [80], ^2^ [33],^3^ [37],^4^ [47], NA-not available.

**Table 6 nanomaterials-11-01988-t006:** Surface area and pore width of various calcium carbonate nanoparticles.

Materials	Names	Surface Area (m^2^/g)	Pore Width (nm)
Blood cockle shells derived calcium carbonate nanoparticles ^1^	CSCaCO_3_NP *****	10.7	5.2
	GEF-CSCaCO_3_NP *****	8.3	5.4
	PTXL-CSCaCO_3_NP *****	9.4	5.6
	GEF-PTXL-CSCaCO_3_NP *****	9.9	5.2
Cockle shells derived CaCO_3_ NP ^2^	CS-CaCO_3_NP *****	6.18 ± 0.65	4.48
Cockle shells derived CaCO_3_ NP ^3^	CaCO_3_ NP *****	6.95	7.12
	DTX-CaCO_3_NP *****	38.73	4.03
Cockle shells derived CaCO_3_ NP ^4^	CaCO_3_ NP (in various solvents) **^+^**	0.96–26	1.8–7.58
Calcium carbonate nanoparticles ^5^	N-Cal **^+^**	15.8	NA
Calcium carbonate nanoparticles ^6^	Amorphous CaCO_3_ NP **^#^**	~350	8–9

^1^ Current study, ^2^ [33], ^3^ [37], ^4^ [53],^5^ [82], ^6^ [112], ***** Aragonite, **^+^** Calcite, **^#^** Amorphous. CSCaCO_3_NP, Blood cockle shell derived CaCO_3_ nanoparticles; GEF-CSCaCO_3_NP, GEF loaded Blood cockle shell derived CaCO_3_ nanoparticles; PTXL-CSCaCO_3_NP, PTXL loaded Blood cockle shell derived CaCO_3_ nanoparticles; GEF-PTXL-CSCaCO_3_NP, GEF and PTXL loaded Blood cockle shell derived CaCO_3_ nanoparticles; DTX-CaCO_3_NP, Docetaxel loaded cockle shell derived CaCO_3_ nanoparticles; N-Cal, Nano calcium carbonate; NA, not available.

## Data Availability

All data generated or analyzed during this research are included in this published article (and its Appendix A) in the form of figures or tables. The datasets generated during and/or analyzed in this current study are available from the corresponding author on a reasonable request.

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
