# Peer review of "Synthesis and Characterization of Gefitinib and Paclitaxel Mono and Dual Drug-Loaded Blood Cockle Shells (Anadara granosa)-Derived Aragonite CaCO3 Nanoparticles"

_nanomaterials, 2021, doi:10.3390/nano11081988_

Round 1

Reviewer 1 Report

The paper by S Chemmalar et al “Synthesis and Characterization of Gefitinib and Paclitaxel Mono and Dual Drug-Loaded Blood Cockle Shells (Anadara Granosa) Derived Aragonite CaCO3 Nanoparticles” deals with rather important topic, both for medicine and environment. The method of drug binding to the surface of nanoparticles seems to be very promising in the struggle against cancer and not only. The paper contains much of new data on the preparation method and properties of calcium carbonate nanoparticles. By a broad number of different methods including TEM, XRD, FTIR spectroscopy, chromatography, specific surface and porosity measurements, etc., the authors have studied the properties of the prepared nanoparticles and supported species, and have demonstrated the advantages of their approach. The results are illustrated by a reasonable number of figures and tables, the conclusions are mostly clear and convincing. English is good enough. The paper could be published in “Nanomaterials” but after serious revision.

1) Throughout the whole paper the same mistake is repeated. Values of measurements, normally repeated thrice, are expressed as mean +/- standard deviation. That is correct, but then the standard deviation should be round up to one meaningful digit. As an exception, if the standard deviation is between 1 and 2 it is acceptable to write, say, ± 1.5. Instead of 83.90 ± 28.19 should be 80 ± 30. Sorry, but it is foolish to write four digits if the first one is not certain. The last meaningful digit in the value must be the same order of magnitude as the standard deviation. You cannot write (as in Table 3):         0.3 ± 0.02! Either 0.30 ± 0.02 or 0.3 ± 0.1.  One should not overestimate the precision. BET model is approximate, instead of 10.68 m2/g better is to write not more than 10.7.

2) Even in the summary it is stated that the drugs are covalently bound to CSCaCO3NP, and this is indicated by FTIR spectroscopy. FTIR data show only that spectrum of carbonate ions are not changed, and modifications in the spectra of drugs do not evidence for the covalent nature of the bond. Why not to say simply that it is strongly bound? In fact, we cannot imagine how the ionic particle can form covalent bond with some molecule. Is it a coordinate complex with calcium? But then it is not covalent, or the drug reacts with carbonate ions?

3) In figure 5 and fig.A3  the spectra are presented in % of transmittance. Then from the scale it should be clear, where is 0 and where 100% .  It is completely not the same if the shown region is from 90 to 100%, or from 0 to 10%. For example, one can mark at the scale 80 and 100% and write in the legend that this corresponds to the upper curve, while the others are shifted down for clarity.  

We have also several minor comments:

4) line 117:  will help achieve…   maybe better:  will help to achieve 

5) line 293:  averaging 64 scans/second.   Apparently should be:  averaging 64 scans.

6) line 399: Ca/C/O are in the molar ratio of 1:1:3   Maybe,better: Ca/C/O are in the molar ratio close to 1:1:3.   Figures in the table are far enough from such an exact ratio.

7) lines 443-444: band exhibited at 1445 cm-1 is attributed to the C-O stretching band. Other peaks at 1084 and 856 cm-1 are attributed to CO32- in the calcium carbonate's molecular structure.
no need to separate the 1445 cm-1 band from other bands of carbonates. Later it is correctly shown in Table 5.

8) Not all the references are presented in the same standard. In paper titles either all the words capitalized, even the prepositions (ref 118), or not. Chemical formulae should be checked up (ref. 60: Fe 3 O 4)   

Author Response

Response to Reviewer 1

General Comments

The paper by S Chemmalar et al “Synthesis and Characterization of Gefitinib and Paclitaxel Mono and Dual Drug-Loaded Blood Cockle Shells (Anadara Granosa) Derived Aragonite CaCO3 Nanoparticles” deals with rather important topic, both for medicine and environment. The method of drug binding to the surface of nanoparticles seems to be very promising in the struggle against cancer and not only. The paper contains much of new data on the preparation method and properties of calcium carbonate nanoparticles. By a broad number of different methods including TEM, XRD, FTIR spectroscopy, chromatography, specific surface and porosity measurements, etc., the authors have studied the properties of the prepared nanoparticles and supported species, and have demonstrated the advantages of their approach. The results are illustrated by a reasonable number of figures and tables, the conclusions are mostly clear and convincing. English is good enough. The paper could be published in “Nanomaterials” but after serious revision.

Major points

Major comment 1. Throughout the whole paper the same mistake is repeated. Values of measurements, normally repeated thrice, are expressed as mean +/- standard deviation. That is correct, but then the standard deviation should be round up to one meaningful digit. As an exception, if the standard deviation is between 1 and 2 it is acceptable to write, say, ± 1.5. Instead of 83.90 ± 28.19 should be 80 ± 30. Sorry, but it is foolish to write four digits if the first one is not certain. The last meaningful digit in the value must be the same order of magnitude as the standard deviation. You cannot write (as in Table 3):   0.3 ± 0.02! Either 0.30 ± 0.02 or 0.3 ± 0.1.  One should not overestimate the precision. BET model is approximate, instead of 10.68 m2/g better is to write not more than 10.7.

RESPONSE:

Thank you very much for the comments. We have corrected the Mean +/- Standard Deviation data throughout the paper as per the comments and rounded them off to one decimal point.

Major comment 2. Even in the summary it is stated that the drugs are covalently bound to CSCaCO3NP, and this is indicated by FTIR spectroscopy. FTIR data show only that spectrum of carbonate ions are not changed, and modifications in the spectra of drugs do not evidence for the covalent nature of the bond. Why not to say simply that it is strongly bound? In fact, we cannot imagine how the ionic particle can form covalent bond with some molecule. Is it a coordinate complex with calcium? But then it is not covalent, or the drug reacts with carbonate ions?

RESPONSE:

Thank you very much for the valuable feedback. The drugs do not react to the calcium carbonate because the spectra of the drugs from the release studies show that there are no changes in the peaks derived from the UV-Vis Spectra observed from the supernatant of the release media.  FTIR spectra of the drug loaded nanoparticles show a few new peaks suggestive of bonding between the drug and the nanoparticles. And we have removed the statement indicating that the drugs are covalently bound but have stated that they are bound to the nanoparticles. [Pg,1 Ln:33 ], [Pg,21 Ln: 1164] [Pg,24 Ln: 1317]

Major comment 3. In figure 5 and fig.A3  the spectra are presented in % of transmittance. Then from the scale it should be clear, where is 0 and where 100% .  It is completely not the same if the shown region is from 90 to 100%, or from 0 to 10%. For example, one can mark at the scale 80 and 100% and write in the legend that this corresponds to the upper curve, while the others are shifted down for clarity.  

RESPONSE:

Thank you very much for pointing this out. We have added the corresponding Transmittance values for each Nanoparticle in the caption of Figure 5 [Pg 13, Ln: 617-624].  We have added the Transmittance value in the y-axis of Figure A3 a and b.

Minor points

Minor comment 1. line 117:  will help achieve…   maybe better:  will help to achieve 

RESPONSE:

Corrected [Pg 3, Ln:145].

Minor comment 2.  line 293:  averaging 64 scans/second.   Apparently should be:  averaging 64 scans.

RESPONSE:

Corrected [Pg 7, Ln: 330].

Minor comment 3.  line 399: Ca/C/O are in the molar ratio of 1:1:3   Maybe, better: Ca/C/O are in the molar ratio close to 1:1:3.   Figures in the table are far enough from such an exact ratio.

RESPONSE:

Corrected [Pg 10, Ln:456 ].

Minor comment 4. lines 443-444: band exhibited at 1445 cm-1 is attributed to the C-O stretching band. Other peaks at 1084 and 856 cm-1 are attributed to CO32- in the calcium carbonate's molecular structure.no need to separate the 1445 cm-1 band from other bands of carbonates. Later it is correctly shown in Table 5.

RESPONSE:

Corrected [Pg 13, Ln: 603-604].

Minor comment 4. Not all the references are presented in the same standard. In paper titles either all the words capitalized, even the prepositions (ref 118), or not. Chemical formulae should be checked up (ref. 60: Fe 3 O 4)

RESPONSE:

Thank you very much for pointing this out. The references have been edited to the format of the Journal. All the chemical formula in the titles have been corrected.

Reviewer 2 Report

The authors showed very interesting content for extracting and applying CaCO3 NPs using an environmentally friendly method, and the results are also systematic.
For this reason, this reviewer actively recommends the publication of this paper.

minor typo

  1. I recommend that you modify the column in Figure 3. 
  2. It is recommended to increase the scale bar of the SEM image in Figure 3.

Yours sincerely,

Onew of reviewers

Author Response

General Comments

The authors showed very interesting content for extracting and applying CaCO3 NPs using an environmentally friendly method, and the results are also systematic.
For this reason, this reviewer actively recommends the publication of this paper.

Minor points

Minor comment 1. I recommend that you modify the column in Figure 3. 

RESPONSE:

Thank you very much for the comment.  We have modified the columns and have incorporated the scanning electron micrographs into the corresponding BET figures and merged them into one Figure 3.

Minor comment 2. It is recommended to increase the scale bar of the SEM image in Figure 3.

RESPONSE:

We have increased the font size of the scale bar of the SEM images in Figure 3.

This manuscript is a resubmission of an earlier submission. The following is a list of the peer review reports and author responses from that submission.

Round 1

Reviewer 1 Report

Review attached

Author Response

General Comments

Razak et al. described in this study the synthesis and characterization of Gefitinib and Paclitaxel mono and dual drug-loaded blood cockle shells derived aragonite CaCO3 nanoparticles as drug delivery systems. Generally, the experiments were properly designed and performed. This study presents some useful data for the use of CaCO3-based nanoparticles for anticancer drug delivery, which merits the publication in Nanomaterials. However, the following points should be considered carefully and fully addressed to improve the scientific quality of this paper prior to acceptance. The overall recommendation is a minor revision.

RESPONSE:

Thank you very much for the comments. We have tried to refine the English and rectify the spelling errors in the manuscript. The conclusions section has been improved [Pg25, Ln: 954-971].

Major points

Major comment 1. The innovation and uniqueness of this study relative to the numerous published papers with a similar research subject on the use of CaCO3-based nanoparticles for anticancer drug delivery remain unclear, which should be highlighted clearly in the abstract and introduction sections.

RESPONSE:

Thank you very much for the valuable comments. The innovation and uniqueness of the current study have been elaborated in the introduction section [Pg 3, Ln:143 - 173]. We have highlighted the knowledge gap and innovation of this study at the beginning of the abstract [Pg1, Ln: 22- 25].

Major comment 2. In vitro drug release profiles should be investigated and reported.

RESPONSE:

We accede to the idea of investigating and reporting the in vitro drug release profiles of GEF, PTXL and GEF-PTXL from the GEF-CSCaCO3NP and PTXL-CSCaCO3NP and dual drug-loaded GEF-PTXL-CSCaCO3NP. However, we apologize that we could not publish that data in this current manuscript, since we are still carrying out the experiments and we plan to publish the data in a separate article that will cover an extensive study on the pH-dependent solubility profiles of aragonite CSCaCO3NP, release profiles of calcium from the aragonite CSCaCO3NP pertaining to its pH-dependent solubility and the drug release profiles of drug-loaded nanoparticles in various pH of 7.4, 6.5 and 5.6. We wanted to preserve the essence of this manuscript by focusing only on certain physicochemical characterization parameters of the newly synthesized CSCaCO3NP, GEF-CSCaCO3NP, PTXL-CSCaCO3NP, and dual drug-loaded GEF-PTXL-CSCaCO3NP. We are afraid that if we add the drug release profiles it would move the focus away from the important data already in the manuscript.

Major comment 3. The rationality for simultaneous loading of Gefitinib and Paclitaxel should be provided.

RESPONSE:

Thank you very much for pointing this out. The rationale behind simultaneous loading of Gefitinib and Paclitaxel in a single system has been extensively covered in the introduction [Pg2, Ln: 88- 108]; [Pg3, Ln:143-17].

Reviewer 2 Report

The authors have reported a promising nanomedicine using biocompatible calcium carbonate nanoparticles and encapsulation of drugs. Their idea is fantastic to develop magic pills for cancer treatment while recycling shell waste. The conclusions are well supported by the experimental data and comparisons with works from fellow researchers.

I would recommend it to be published after minor revision. It would be of interests for the audience and beneficial for future developments.

The concerns raised from observing the SEM images is the aggregation issue. The aggregates are much bigger than the synthesized calcium carbonate nanoparticles. The increase in size might affect application in the terms of uptake, distribution, etc and may cause blockage along the pathway. Could the authors indicate some solutions to prevent aggregation from preparation through to application stage?

The CSCaCO3NPs are described as active targeting nanomaterials. It would be appreciative if the authors elaborate on the targeting path involved in the NPs. For example, as EGFR is present on many normal cells and cancer cells, GEF loaded CSCaCO3NPs might be associated with both types of cells but CSCaCO3NPs dissociate only in the cancer environment and release the loadings, leaving normal cells unaffected by the drug.

In the discussion, the details for encapsulation efficiency of PTXL formulations went missing on Page 15.

In some figures, the graphs/plots are stretched, giving distorted view. The presentation of graphs should be revised.

Author Response

General Comments

The authors have reported a promising nanomedicine using biocompatible calcium carbonate nanoparticles and encapsulation of drugs. Their idea is fantastic to develop magic pills for cancer treatment while recycling shell waste. The conclusions are well supported by the experimental data and comparisons with works from fellow researchers.

I would recommend it to be published after minor revision. It would be of interests for the audience and beneficial for future developments.

RESPONSE:

Thank you very much for the comments. We have tried to refine the English and rectify the spelling errors in the manuscript.

Major points

Major comment 1. The concern raised from observing the SEM images is the aggregation issue. The aggregates are much bigger than the synthesized calcium carbonate nanoparticles. The increase in size might affect the application in the terms of uptake, distribution, etc and may cause blockage along the pathway. Could the authors indicate some solutions to prevent aggregation from preparation through to application stage?

RESPONSE:

Thank you for the comments.  The problem of aggregation has been adequately addressed and a few pre-emptive methods has been described for future research in the discussion section [Pg21, Ln: 717-732].

Major comment 2. The CSCaCO3NPs are described as active targeting nanomaterials. It would be appreciative if the authors elaborate on the targeting path involved in the NPs. For example, as EGFR is present on many normal cells and cancer cells, GEF loaded CSCaCO3NPs might be associated with both types of cells but CSCaCO3NPs dissociate only in the cancer environment and release the loadings, leaving normal cells unaffected by the drug.

RESPONSE:

The calcium carbonate nanoparticles by themselves are not an active targeting nanomaterial.  However, when we surface functionalize with an active targeting drug like Gefitinib (Epidermal growth factor tyrosine kinase inhibitor), it can be called an Active targeting nanomedicine. The mechanism of action of the synthesized CSCaCO3NPs is explained in the discussion section [Pg24,25, Ln: 923-949]. Since we have two more components in the system, the mechanism of action of Gefitinib is also elaborated in the introduction section [Pg 2, Ln: 89-107], and the synergism with PTXL is also clearly explained in the introduction section [Pg 3, Ln: 153-160].

Major comment 3. In the discussion, the details for encapsulation efficiency of PTXL formulations went missing on Page 15.

RESPONSE:

Thank you very much for pointing this out. We have added the missing data for encapsulation efficiency of PTXL in the discussion [Pg 20, Ln: 644-645].

Major comment 4. In some figures, the graphs/plots are stretched, giving distorted view. The presentation of graphs should be revised.

RESPONSE:

Thank you very much for your valuable comments. We have tried our best to improve the quality of the graphs and plots.

Reviewer 3 Report

i could not find a novelty in this paper as i read it multiple times and the references are also given a better presentation of the methods given here.

Author Response

General Comments

I could not find a novelty in this paper as i read it multiple times and the references are also given a better presentation of the methods given here.

RESPONSE:

We deeply apologize for not have written the manuscript clearly and concisely. Thank you very much for pointing that out. We have extensively edited the language and style of the manuscript.

The introduction, results and conclusions sections have been improved overall. The uniqueness and novelty of this current study are elaborated in the introduction [Pg 3, Ln: 150-173].

Round 2

Reviewer 3 Report

I have reviewed the new version of the manuscript too. However, this new edition only includes some literature review and small details added to the paper. the major deficiency of the paper is that the in vitro drug release profiles of GEF, PTXL and GEF-PTXL from the GEF-CSCaCO3NP and PTXL-CSCaCO3NP and dual drug-loaded GEF-PTXL-CSCaCO3NP are missing and this has no single value to the readers and I must reject it.